# Spatial and temporal coordination of Duox/TrpA1/Dh31 and IMD pathways is required for the efficient elimination of pathogenic bacteria in the intestine of *Drosophila* larvae

**Fatima Tleiss[1], Martina Montanari[2], Romane Milleville[2], Olivier Pierre[1], Julien Royet[2]\*, Dani Osman[3]\*, Armel Gallet[1]\*, C Leopold Kurz[2]\***

[1]Université Côte d'Azur, CNRS, INRAE, ISA, Nice, France; [2]Aix-Marseille Université, CNRS, IBDM, Marseille, France; [3]UMR PIMIT (Processus Infectieux en Milieu Insulaire Tropical) CNRS 9192-INSERM 1187-IRD 249-Université de La Réunion, île de La Réunion, France

**\*For correspondence:**
julien.royet@univ-amu.fr (JR);
dani.osman@univ-reunion.fr
(DO);
armel.gallet@inrae.fr (AG);
leopold.kurz@univ-amu.fr (CLK)

**Competing interest:** The authors declare that no competing interests exist.

## eLife Assessment

This article describes a novel mechanism allows *Drosophila* to combat enteric pathogens while also preserving the beneficial indigenous microbiota. The authors provide **compelling** evidence that oral infection of *Drosophila* larvae by pathogenic bacteria activate a valve that traps the intruders in the anterior midgut, allowing them to be killed by antimicrobial peptides. This is an **important** finding revealing a new mechanism of host defense in the gut of insects.

**Abstract** Multiple gut antimicrobial mechanisms are coordinated in space and time to efficiently fight foodborne pathogens. In *Drosophila melanogaster*, production of reactive oxygen species (ROS) and antimicrobial peptides (AMPs) together with intestinal cell renewal play a key role in eliminating gut microbes. A complementary mechanism would be to isolate and treat pathogenic bacteria while allowing colonization by commensals. Using real-time imaging to follow the fate of ingested bacteria, we demonstrate that while commensal *Lactiplantibacillus plantarum* freely circulate within the intestinal lumen, pathogenic strains such as *Erwinia carotovora* or *Bacillus thuringiensis*, are blocked in the anterior midgut where they are rapidly eliminated by antimicrobial peptides. This sequestration of pathogenic bacteria in the anterior midgut requires the Duox enzyme in enterocytes, and both TrpA1 and Dh31 in enteroendocrine cells. Supplementing larval food with hCGRP, the human homolog of Dh31, is sufficient to block the bacteria, suggesting the existence of a conserved mechanism. While the immune deficiency (IMD) pathway is essential for eliminating the trapped bacteria, it is dispensable for the blockage. Genetic manipulations impairing bacterial compartmentalization result in abnormal colonization of posterior midgut regions by pathogenic bacteria. Despite a functional IMD pathway, this ectopic colonization leads to bacterial proliferation and larval death, demonstrating the critical role of bacteria anterior sequestration in larval defense. Our study reveals a temporal orchestration during which pathogenic bacteria, but not innocuous, are confined in the anterior part of the midgut in which they are eliminated in an IMD-pathway-dependent manner.

## Introduction

One of the key avenues by which bacterial pathogens infiltrate a host is through the ingestion of contaminated food. Following the entry of bacteria in the intestine, the host defense mechanisms will operate in a temporal manner, with mechanical and constitutive chemical barriers serving as the first line of defense, followed by inducible mechanisms involving the production of reactive oxygen species (ROS), the transcription, translation, and secretion of antimicrobial peptides (AMPs) as well as inter-organ signaling to cope with possible upcoming stages of infection. This temporality is evident between innate and adaptive immunity, with the former considered the primary defense line that contains and combats the threat while preparing the more subtle adaptive response.

To focus on deciphering innate immune processes, the insect *Drosophila melanogaster* has been widely and successfully used (*Neyen et al., 2014*; *Younes et al., 2020*). This model has made it possible to establish the chronology of the events involved in the defense against pathogenic bacteria. In *Drosophila*, as in all metazoans, a layer made of mucus, completed with a peritrophic membrane in insect midguts, protects the intestine lining from direct contact with pathogens (*Hegedus et al., 2009*; *Lemaitre and Miguel-Aliaga, 2013*; *Pelaseyed et al., 2014*). In adult *Drosophila*, a conserved immune response involving the production of ROS by Duox (dual oxidase) enzyme in enterocytes (ECs) is triggered in the intestine as early as 30 min after ingesting pathogenic bacteria. ROS directly damage bacterial membranes (*Benguettat et al., 2018*; *Ha et al., 2009*; *Ha et al., 2005*; *Lee et al., 2013*) but also exert an indirect effect in adults by triggering visceral spasms through the host detection of ROS mediated by the TrpA1 nociceptor and subsequent secretion of diuretic hormone 31 (Dh31) by enteroendocrine cells (EECs; *Benguettat et al., 2018*; *Du et al., 2016a*). Dh31 then binds its receptor on visceral muscles, triggering contractions that expedite bacterial elimination (*Benguettat et al., 2018*). This pathway seems to be conserved during evolution as TrpA1 is a *Drosophila* homolog of TRP receptors that respond to noxious conditions (*Ogawa et al., 2016*) and Dh31 is the *Drosophila* homolog of the mammalian CGRP (*Guo et al., 2022*; *Nässel and Zandawala, 2019*). In parallel, the immune deficiency (IMD) innate immune pathway is activated following bacterial peptidoglycan detection, leading to the transcription of AMP encoding genes and to the subsequent secretion of the peptides that kill bacteria (*Capo et al., 2019*).

Previous study has explored the dynamics of food transit involving intestinal valves and Dh31 in the context of *Drosophila* larvae (*LaJeunesse et al., 2010*). Additionally, the perturbed circulation of bacteria within the intestine has not been documented in the literature though some specific localization of commensal and pathogenic bacteria along the intestine have been observed in insects (*Basset et al., 2000*; *Bosco-Drayon et al., 2012*; *Lanan et al., 2016*; *Nardi et al., 2016*; *Ramond et al., 2021*; *Yao et al., 2022*). However, the intricate interplay between the sequestration of bacteria in specific gut regions and their subsequent elimination has remained unexplored. In our study, we leveraged a novel real-time experimental system designed specifically for *Drosophila* larvae, enabling us to meticulously trace the journey and ultimate fate of pathogenic bacteria ingested alongside food. This approach has allowed us to shed light on the complex mechanisms underpinning bacterial management within the larval gut, contributing a significant advancement to our understanding of host-pathogen interactions at the intestinal level. We characterized a new mechanism implicated in the blockage and elimination of pathogenic bacteria in the anterior part of the midgut. We demonstrated that this confinement is regulated by the ROS/TrpA1/Dh31 axis. Our results delineate a model in which bacterial trapping arises from ROS production in the intestinal lumen in response to pathogenic bacteria. These ROS compounds interact with TrpA1 in Dh31-expressing EECs located between the anterior and middle midgut, leading to Dh31 secretion and subsequent bacterial compartmentalization, suggesting the closure of a valve, a midgut junction structure proposed by *LaJeunesse et al., 2010*. Interestingly, we found that we can ectopically induce the trapping of fluorescent particles or innocuous bacteria using human CGRP that replaces Dh31. Our findings also highlight the central role of this blockage, which acts first allowing sufficient time for the subsequent eradication of blocked pathogens by the IMD pathway and its downstream effectors. Collectively, our data unravel a finely tuned coordination between the ROS/TrpA1/Dh31 axis and the IMD pathway, enabling an effective bactericidal action of AMPs.

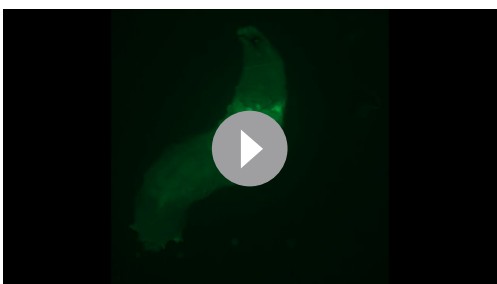

**Video 1.** Fluorescent *Ecc* is blocked in the anterior part of the larval intestine then vanishes. Live imaging during 12 hr of a L3 control larva previously fed 1 hr with a food containing *Ecc* fluorescent bacteria then transferred on a glass slide in a wet chamber. https://doi.org/10.6084/m9.figshare.25018385.v2.
https://elifesciences.org/articles/98716/figures#video1

## Results

### Bacterial confinement in the anterior part of larval intestine

The translucency of *Drosophila* larvae allows for live studies of immune defense components and their coordination in eradicating pathogenic bacteria. In our prior work, we revealed a modified food transit in larvae exposed to the Gram-negative opportunistic bacterium *Erwinia carotovora carotovora* (*Ecc15*), a process that involved TrpA1 (*Keita et al., 2017*).

Using blue food dye, we tracked the presence of food in the intestinal lumen and observed that larval guts were blue without bacterial contaminants, while in the presence of *Ecc15*, they appeared clearer (*Keita et al., 2017*). Such a strategy was already used to delineate whether oral infection by *Pseudomonas entomophila* could modulate food transit (*Liehl et al., 2006*). However, our previous assay with *Ecc15* was limited to 1 hr post-ingestion and using a food dye, not directly moenitoring the fate of the ingested bacteria over the time. We therefore designed a new protocol allowing to film the fate of fluorescent bacteria once ingested by the larvae. Animals are fed for 1 hr with food contaminated with fluorescent bacteria to investigate the localization of these pathogens within the intestinal tract. Following this feeding period, the animals were transferred to a wet chamber devoid of food. This setup allowed us to monitor the positional dynamics of the bacteria within the intestine over time, without further food intake influencing the observations. We principally tested three different fluorescent bacteria: *Ecc15-GFP* (*Ecc*, an opportunistic Gram-negative bacterium), *Bacillus thuringiensis-GFP* (*Bt*, an opportunistic Gram-positive bacterium), and *Lactiplantibacillus plantarum-GFP* (*Lp*, a commensal Gram-positive bacterium). After 1 hr of feeding on contaminated media, *Ecc* and *Bt* were concentrated in the anterior midgut (*Videos 1 and 2*). The location of the bacteria specifically in the anterior part of the intestine following 1 hr exposure was confirmed when individuals from populations were imaged and counted (*Figure 1A*). The consistently defined limits of the area containing bacteria in the anterior part of the gut across our observations strongly indicate the presence of a physical boundary. This reproducibility contrasts with what would be expected if the cessation of food intake was responsible for halting bacterial progression, which would likely result in more variable bacterial distributions within the intestine among individual subjects (*Figure 1A*). Remarkably, with animals fed 1 hr with *Ecc* or *Bt* and transferred in a wet chamber, tracking the GFP signal over time revealed that it remained in the anterior part of the larva and began to fade 6 hr after ingestion (*Videos 1 and 2*; *Figure 1—figure supplement 1A and B*). This fading suggests the elimination of the bacteria, while the larva continued to exhibit active movements. This pattern was observed for both *Ecc* and *Bt*. Interestingly, the blockage is reversible and may depend upon the presence or status of the trapped bacteria. Indeed, if fluorescent dextran polymers are co incubated with *Ecc*, both bacteria and dextran are blocked in the anterior part of the intestine and following bacteria-associated fluorescence disappearance, the dextran is released in the posterior part (*Video 3*). However, unlike the opportunistic bacteria *Ecc* and *Bt*, a food mixture contaminated with *Lp* led to the bacteria being present in the posterior compartment of the larval midgut after 1 hr of feeding and remained there throughout the 16 hr duration of our observation (*Video 4*,

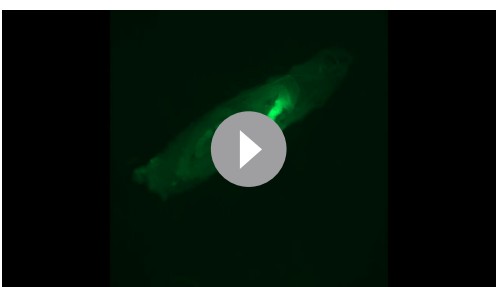

**Video 2.** Fluorescent *Bt* is blocked in the anterior part of the larval intestine then vanishes. Live imaging during 12 hr of a L3 control larva previously fed 1 hr with a food containing *Bt* fluorescent bacteria then transferred on a glass slide in a wet chamber. https://doi.org/10.6084/m9.figshare.25018427.v2.
https://elifesciences.org/articles/98716/figures#video2

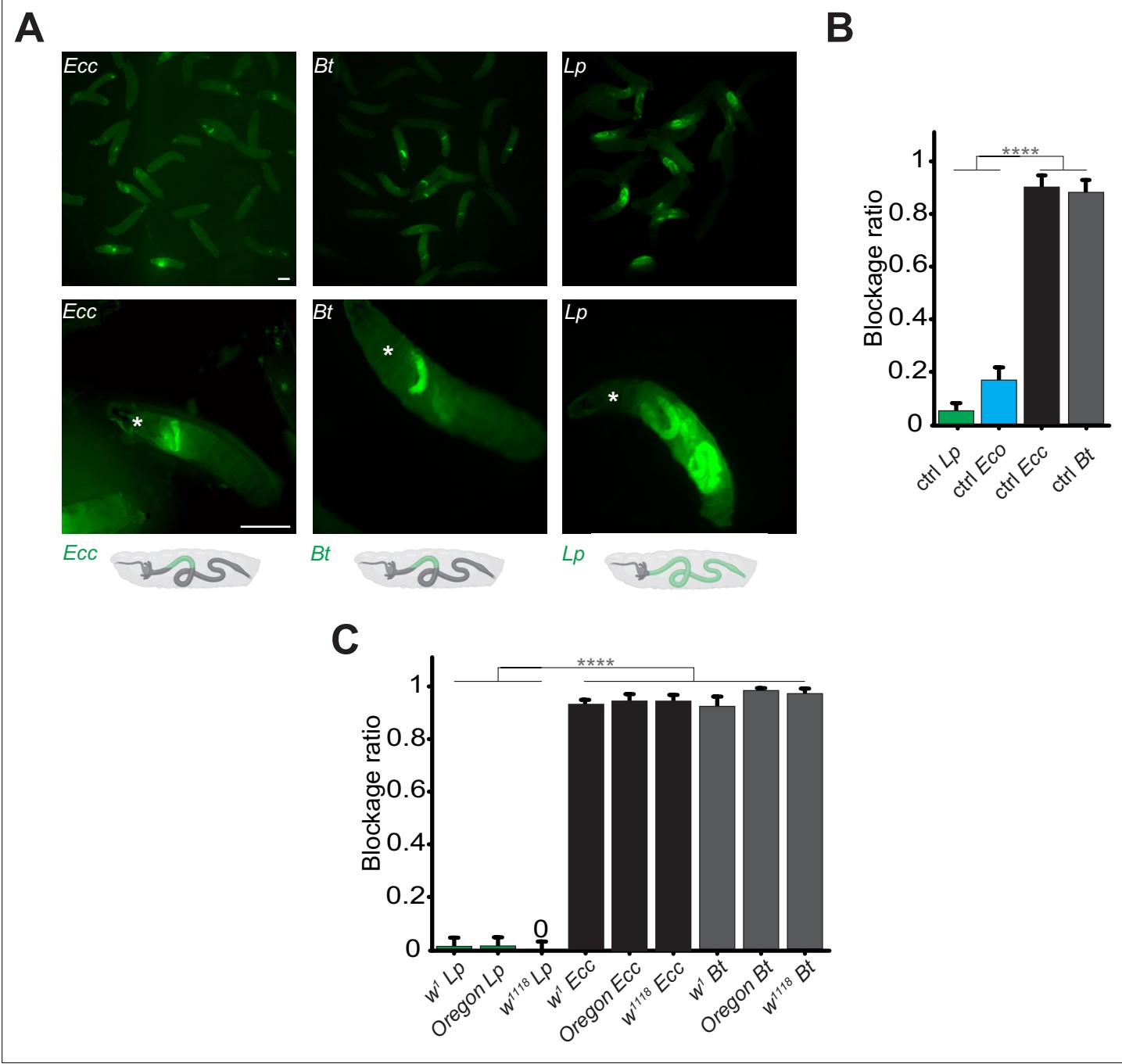

**Figure 1.** Contrary to *Lp or Eco*, *Ecc* or *Bt* bacteria are found exclusively in the anterior part of the gut. (**A**) Pictures to illustrate the position of the green fluorescence of control (CantonS) L3 stage larvae as a group (upper panel) or individual (lower panel) after having been fed 1 hr with a media containing yeast and GFP-producing bacteria (*Ecc* or *Bt* or *Lp*). The white asterisk indicates the anterior part of the animal. The white arrow indicates the posterior limit of the area containing the fluorescent bacteria. Below the pictures are schematics representing larvae, their gut, and the relative position of the GFP-producing bacteria in green. Scale bar is 1 mm. (**B**) Graphic representing the blockage ratio for CantonS (ctrl) L3 larvae exposed during 1 hr to a mixture composed of yeast and fluorescent bacteria: *Lp* or *Escherichia coli* (*Eco*) or *Ecc* or *Bt*. The ratio of control larvae with a distinguishable green fluorescence only in the upper part of the intestine, considered as blocked bacteria, is represented. The ratio is calculated as: x larvae with bacteria exclusively in the anterior part of the gut / (x larvae with bacteria exclusively in the anterior part of the gut +y larvae with bacteria all along the gut). Larvae with no distinguishable fluorescence were considered as non-eaters and discarded from the quantifications. The ratio of larvae with no distinguishable fluorescence was not influenced by the different conditions we tested. Shown is the average blockage ratio with a 95% confidence interval from at least three independent assays with at least 30 animals per condition and trial. **** indicates p<0.0001, Fisher exact t-test. See the source data file for details. (**C**) Blockage ratio for L3 larvae of $w^1$, *Oregon* or $w^{1118}$ isogenized genotypes fed during various times with a mixture

*Figure 1 continued on next page*

*Figure 1 continued*

combining yeast with *Lp*, *Ecc*, or *Bt*. Shown is the average blockage ratio with a 95% confidence interval from at least three independent assays with at least 100 animals in total. The 0 symbol indicates an absence of blockage, **** indicates p<0.0001, Fisher exact t-test. See the source data file for details.

The online version of this article includes the following figure supplement(s) for figure 1:

**Figure supplement 1.** *Ecc* and *Bt* are blocked in the anterior part of the intestine and disappear while *Lp* transits to the posterior part and remains.

**Figure supplement 2.** Contrary to *Lp*, *Ecc,* or *Bt* bacteria are blocked in the anterior part of the gut.

*Figure 1A* and *Figure 1—figure supplement 1C*; *Storelli et al., 2018*). To better characterize the blockage phenomenon, we counted the ratio of larvae with trapped GFP-bacteria 1 hr post feeding. Importantly, in all our experiments, approximately 20% of larvae displayed an absence of fluorescent bacteria within the intestinal lumen. Notably, this proportion remained consistent across different bacterial strains used to contaminate the food, indicating that the specific compartmentalization of pathogenic bacteria we observed is not related to a global food-intake cessation. Instead, these findings suggest the deployment of a targeted sequestration mechanism. Focusing on animals containing fluorescent bacteria, we found that while less than 10% of the larvae had *Lp* blocked in the anterior part of the intestine, more than 80% of the larvae had *Ecc* and *Bt* bacteria localized in the anterior part of the midgut (*Figure 1A and B*). We completed our observation of the posterior location of non-pathogenic bacteria with another harmless species, the OP50 strain of *Escherichia coli* (*Eco*). Less than 20% of the animals fed a mixture containing *Eco* showed a blocking phenotype. The portion of the intestine containing the fluorescent bacteria is delimited posteriorly by an extensive turn, a region that was suggested to act like a valve (*LaJeunesse et al., 2010*). Importantly, the blockage of pathogenic bacteria (*Ecc* and *Bt*) in the anterior part of the intestine was confirmed using *D. melanogaster* larvae with different genetic backgrounds (*Figure 1C*). To confirm our findings, we dissected the intestines of larvae that had ingested the bacteria. Our analysis confirmed that *Ecc* and *Bt* were predominantly located in the anterior part of the intestines, whereas *Lp* was not, supporting our initial observations (*Figure 1—figure supplement 2*). Our data collectively indicate that pathogenic bacteria, such as *Ecc* and *Bt*, are spatially confined to the anterior part of the larval midgut before their disappearance. In contrast, the commensal bacterium *Lp* or the

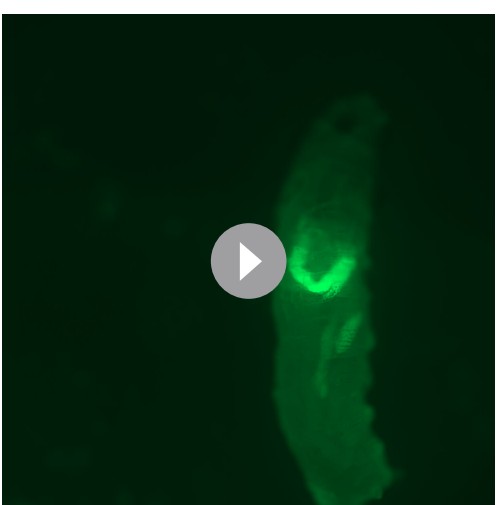

**Video 3.** Dextran-TexasRed is concomitantly blocked Fluorescent *Bt* in the anterior part of the larval intestine then released posteriorly following bacterial clearance. Live imaging during 6 hr of a L3 control larva previously fed 1 hr with a food containing *Bt* fluorescent bacteria and Dextran-Texas-Red then transferred on a glass slide in a wet chamber. The anterior part of the larvae is at the bottom right. https://doi.org/10.6084/m9.figshare.26355076.v1.

https://elifesciences.org/articles/98716/figures#video3

**Video 4.** Fluorescent *Lp* is not blocked in the anterior part of the larval intestine and persists in the posterior midgut. Live imaging during 10 hr of a L3 control larva previously fed 1 hr with a food containing *Lp* fluorescent bacteria then transferred on a glass slide in a wet chamber. https://doi.org/10.6084/m9.figshare.25018442.v2.

https://elifesciences.org/articles/98716/figures#video4

innocuous *Eco* line we used are distributed throughout the midgut, persisting principally in the posterior part.

## The anterior intestinal confinement of pathogenic bacteria is dose-specific and occurs rapidly

We hypothesized that the bacterial localization specifically in the anterior part of the larval intestine was an active host response and might be dependent on the bacterial dose. To test this, we exposed larvae to varying concentrations of *Ecc* and *Bt* and measured the blockage ratio. We found that a concentration of $2.10^9$ *Ecc* bacteria per ml was sufficient to induce the bacterial compartmentalization, whereas for *Bt*, a concentration of $4.10^{10}$ *Bt* bacteria per ml was required (*Figure 2A*). Interestingly, following over time the intoxication of control larvae with a *Bt* concentration of $1.10^{10}$ bacteria per ml instead of $4.10^{10}$ *Bt* bacteria per ml revealed that bacteria were not sequestered in the anterior part of the intestine but reached the posterior part (*Video 5*). Nonetheless, the fluorescence remained weak likely due to the low amount of bacteria and we could even observed an excretion of bacteria. For all subsequent experiments, we used $4.10^{10}$ bacteria per ml.

In previous contamination assays, we arbitrary used a 1 hr time point to assess the phenotype. However, observations of bacterial blockage occurring within minutes suggested that this response does not require de novo protein synthesis by the host. Shorter exposure times revealed that *Ecc* was blocked in the anterior part of the intestine within 15 min, while *Bt* showed a similar pattern beginning at 15 min and completing by 30 min (*Figure 2B and C*). Based on these results, we defined 1 hr as our standard exposure time of larvae with bacterial contaminated food. In addition, we performed longer continuous feeding with *Ecc* (20 hr) to assay whether the blockage would still be present. We observed that while most of the larvae continuously exposed to the contaminated mixture tend to flee and do not contain fluorescent bacteria in their intestine, the ones with detectable fluorescence were blocking the bacteria (*Figure 2—figure supplement 1*).

Our assays typically involved groups of approximately 50 larvae to observe population-level phenomena. Recent studies have suggested that larval behavior can be influenced by group dynamics (*Dombrovski et al., 2019*; *Dombrovski et al., 2017*; *Louis and de Polavieja, 2017*; *Mast et al., 2014*). To determine whether group size affects the bacterial confinement, we exposed groups of varying sizes to *Bt* or *Ecc* and measured the blockage ratio. The phenomenon proved robust even with a single larva exposed to contaminated food, indicating that the response was not influenced by group size under our experimental conditions (*Figure 2D*). Based on these findings, for all subsequent experiments, we standardized the conditions using $4.10^{10}$ bacteria per ml, a 1 hr exposure time, and groups of at least 20 larvae.

## Host TrpA1 and Dh31 are crucial for the blockage phenotype

In our previous study reporting an altered food transit for larvae exposed to a food contaminated with *Ecc* (*Keita et al., 2017*), we identified the gene *TrpA1* as essential for the host response. The TrpA1 protein, a member of the TRP channel family, facilitates Ca2 +entry into cells at temperatures over 25 °C or upon exposure to chemicals such as ROS (*Du et al., 2016b*; *Gu et al., 2019*; *Guntur et al., 2015*). This channel, involved in nociception (*Lapointe and Altier, 2011*), has been linked to intestinal muscle activity in adult *Drosophila* following *Ecc* exposure (*Du et al., 2016a*) and in response to ROS production by the host (*Benguettat et al., 2018*). We examined the role of TrpA1 during the larval response to contaminated food using *TrpA1[1]* homozygous viable mutant in the experimental setup with fluorescent *Lp*, *Ecc*, or *Bt* bacteria. In these mutant larvae, while the localization of *Lp* in the posterior midgut remained unchanged, *Ecc* and *Bt* were found in the posterior midgut following a 1 hr intoxication and the fluorescence does not fade (*Figure 3A and B*, *Video 6* and *Figure 3—figure supplement 1A*). This observation underscores that TrpA1 is necessary for the blockage of pathogenic bacteria in the anterior midgut and suggests that the disappearance of the fluorescence observed for pathogenic bacteria within the intestine of control animals may be due to bacteria being killed rather than fading of GFP or plasmid loss. Considering previous reports on visceral contraction in adult *Drosophila* linked to ROS production, detection by TrpA1, and Dh31 secretion to expel bacteria (*Benguettat et al., 2018*; *Du et al., 2016a*), we wondered whether a similar ROS/TrpA1/Dh31 signaling axis is necessary in larvae to block pathogenic bacteria in the anterior midgut. Indeed, the bacterial blockage we observed might involve muscle contractions related to food contamination.

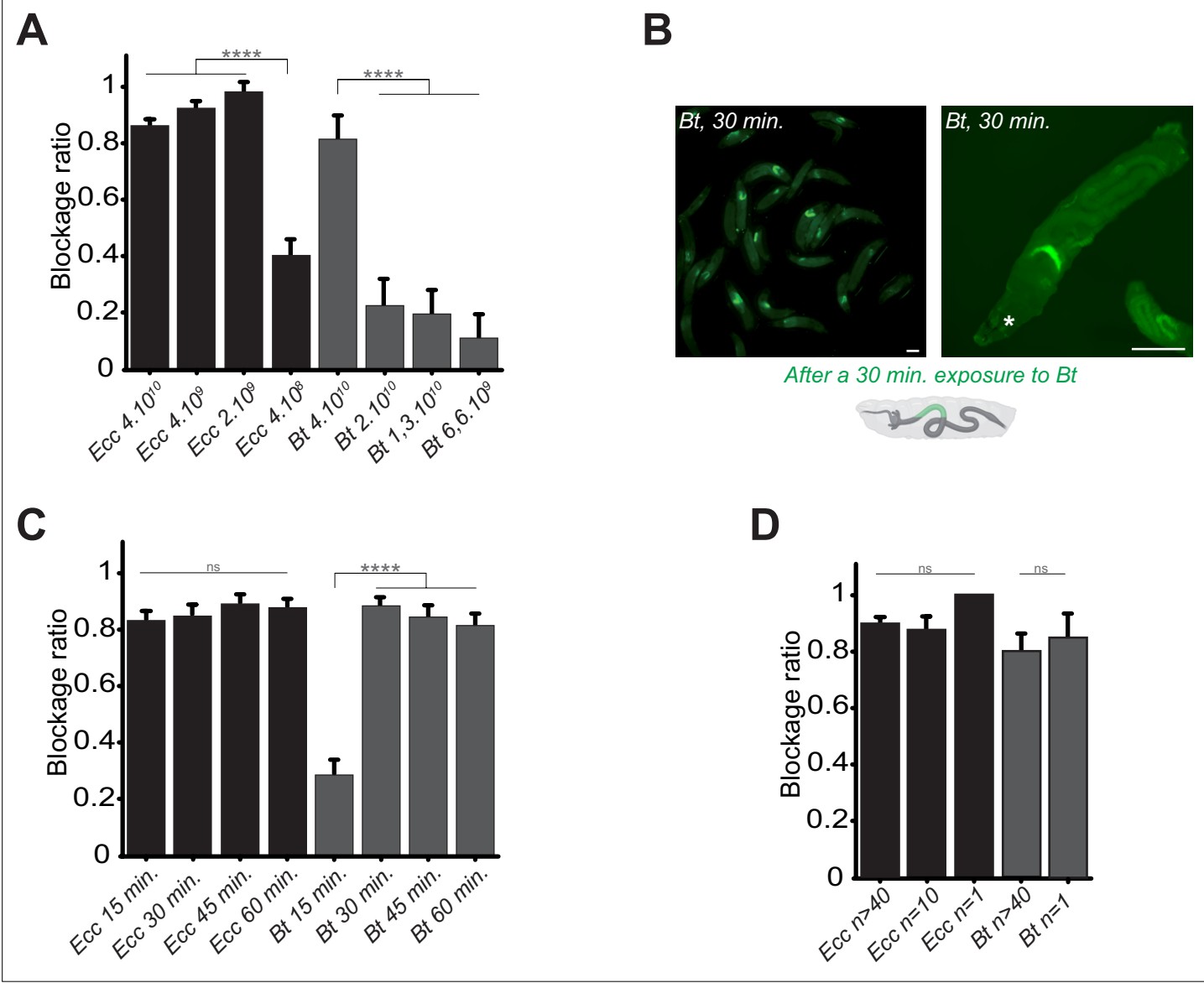

**Figure 2.** Bacterial blockage is dose-dependent, occurs in less than 30 min, and does not involve a group effect. (**A**) Blockage ratio for control L3 larvae fed 1 hr with a mixture combining yeast with different concentrations of fluorescent *Ecc* or *Bt*, concentrations are in number of bacteria per ml. Shown is the average blockage ratio with a 95% confidence interval from at least three independent assays with at least 18 animals per condition and trial. **** indicates p<0.0001, Fisher exact t-test. See the source data file for details. (**B**) Representative images of control larvae fed during 30 min. with *Bt*. Scale bar is 1 mm. (**C**) Blockage ratio for control L3 larvae fed during various times with a mixture combining yeast with *Ecc* or *Bt*. Shown is the average blockage ratio with a 95% confidence interval from at least three independent assays with at least 20 animals per condition and trial. ns indicates values with differences not statistically significant, **** indicates p<0.0001, Fisher exact t-test. See the source data file for details. (**D**) Blockage ratio for control L3 larvae fed 1 hr as individual animals or as groups of 10 or >40 with a mixture combining yeast with a constant concentration of *Ecc* or *Bt* ($4.10^{10}$ bacteria per ml). Shown is the average blockage ratio with a 95% confidence interval from at least three independent assays with the exact number of animals indicated per condition and trial. ns indicates values with differences not statistically significant, Fischer exact t-test. See the source data file for details.

The online version of this article includes the following figure supplement(s) for figure 2:

**Figure supplement 1.** *Bt* is blocked anteriorly after 20 hr of continuous feeding.

Thus, we tested the blockage ratio of *Dh31^{KG09001}* homozygous viable mutants exposed to *Lp*, *Bt*, or *Ecc* bacteria. In contrast to control animals, where fewer than 20% of larvae exhibited *Ecc* or *Bt* in the posterior section of the intestine, for more than 60% of *Dh31^{KG09001}* mutant larvae, the pathogenic bacteria *Bt* or *Ecc* were localized in the posterior midgut confirming Dh31's crucial role in this mechanism

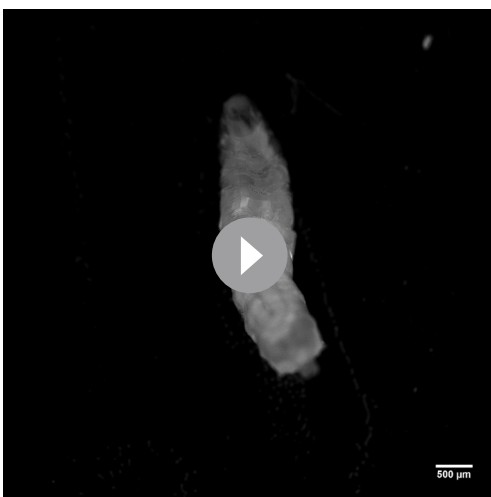

**Video 5.** *Bt* at a low concentration is not blocked in the anterior part of the larval intestine, persists in the posterior midgut and the larva does not die. Live imaging during 10 hr of a L3 control larva previously fed 1 hr with a food containing *Bt* fluorescent bacteria (1.10¹⁰ bacteria per ml instead of 4.10¹⁰) then transferred on a glass slide in a wet chamber. https://doi.org/10.6084/m9.figshare.26355196.v1.
https://elifesciences.org/articles/98716/figures#video5

(*Figure 3A and B*, *Video 7* and *Figure 3—figure supplement 1B*). The localization of *Lp* was not affected (*Figure 3B*).

Given the possibility that the compartmentalization of pathogens we observed might be driven by bacterial virulence factors, we aimed to further investigate the underlying mechanism. To discern whether this phenomenon was specific to pathogenic bacteria or could extend to non-pathogenic entities, we conducted experiments to determine if abiotic particles or non-virulent bacteria could also be ectopically sequestered within the gut. The human Calcitonin Gene-Related Peptide (hCGRP) is the functional homolog of Dh31. Indeed, hCGRP has been shown to promote visceral muscle contractions in adult flies (*Benguettat et al., 2018*). To investigate whether the bacterial confinement could be triggered without pathogenic bacteria in response to a hormone, we exposed larvae to either *Lp* or Dextran-FITC in the presence of hCGRP. While *Lp* and Dextran-FITC were normally distributed throughout the midgut, adding hCGRP to the food induced a significant blockage (*Figure 3B and C* and *Video 8*). This observation implies that the confinement of bacteria within the gut is not solely a direct consequence of pathogen toxicity. Intriguingly, despite the larvae being housed in a wet chamber where no further food intake occurs (which could otherwise push the bacteria towards the posterior gut), we noted that *Lp* initially sequestered in the anterior midgut following the administration of hCGRP began to be released in the posterior part after 6 hr (*Video 8* and *Figure 3—figure supplement 1C*). This is suggestive of a dynamic and reversible phenomenon, likely linked to Dh31/hCGRP hormone metabolization. This demonstrates that the process could be triggered independently of the bacteria by a hormone, independently of food-intake per se, highlighting it as an active host mechanism that involves a specific portion of the gut capable of limiting the circulation of the bacteria.

## Duox in enterocytes and Dh31 in Pros+ cells control pathogenic bacteria blockage

The activation of TrpA1, potentially leading to the release of Dh31 (*Belinskaia et al., 2023*; *Kondo et al., 2010*; *Kunst et al., 2014*), could be a consequence of ROS production in the host larval midgut in response to *Ecc* or *Bt*. The larval intestine comprises two main cell populations: ECs and EECs. In adult *Drosophila*, Dh31 is stored in EECs and is secreted in response to TrpA1 activation, a process well documented in literature (*Benguettat et al., 2018*; *Chen et al., 2016*; *Veenstra et al., 2008*). The role of ROS in the immunity of adult *Drosophila* intestine following infection has also been reported (*Chakrabarti et al., 2012*; *Ha et al., 2005*; *Lee et al., 2013*; *Ryu et al., 2006*). To investigate the involvement of ROS in the larval bacterial confinement, we focused on Duox, the primary enzyme responsible for luminal ROS production. We spatially silenced *Duox* expression using RNA interference (*UAS-Duox_IR*) driven by an ubiquitous driver (Da-Gal4), a driver specific of ECs (Mex-Gal4) or Pros-Gal4, a construction driving expression in EECs and subsets of neuroblasts. In parallel, using the same set of drivers, we tested the effects of cell-autonomous *Dh31* silencing using RNAi (*UAS-Dh31_IR*). Upon exposing these larvae to a food mixture contaminated with fluorescent *Ecc* or *Bt*, we assessed the blockage ratio. Our results indicate that in larvae, Duox is essential in ECs for the blockage of both *Ecc* and *Bt* (*Figure 3D*). Furthermore, silencing *Dh31* in EECs not only confirmed the mutant phenotype but also indicated that Dh31, necessary for the confinement, is required for the

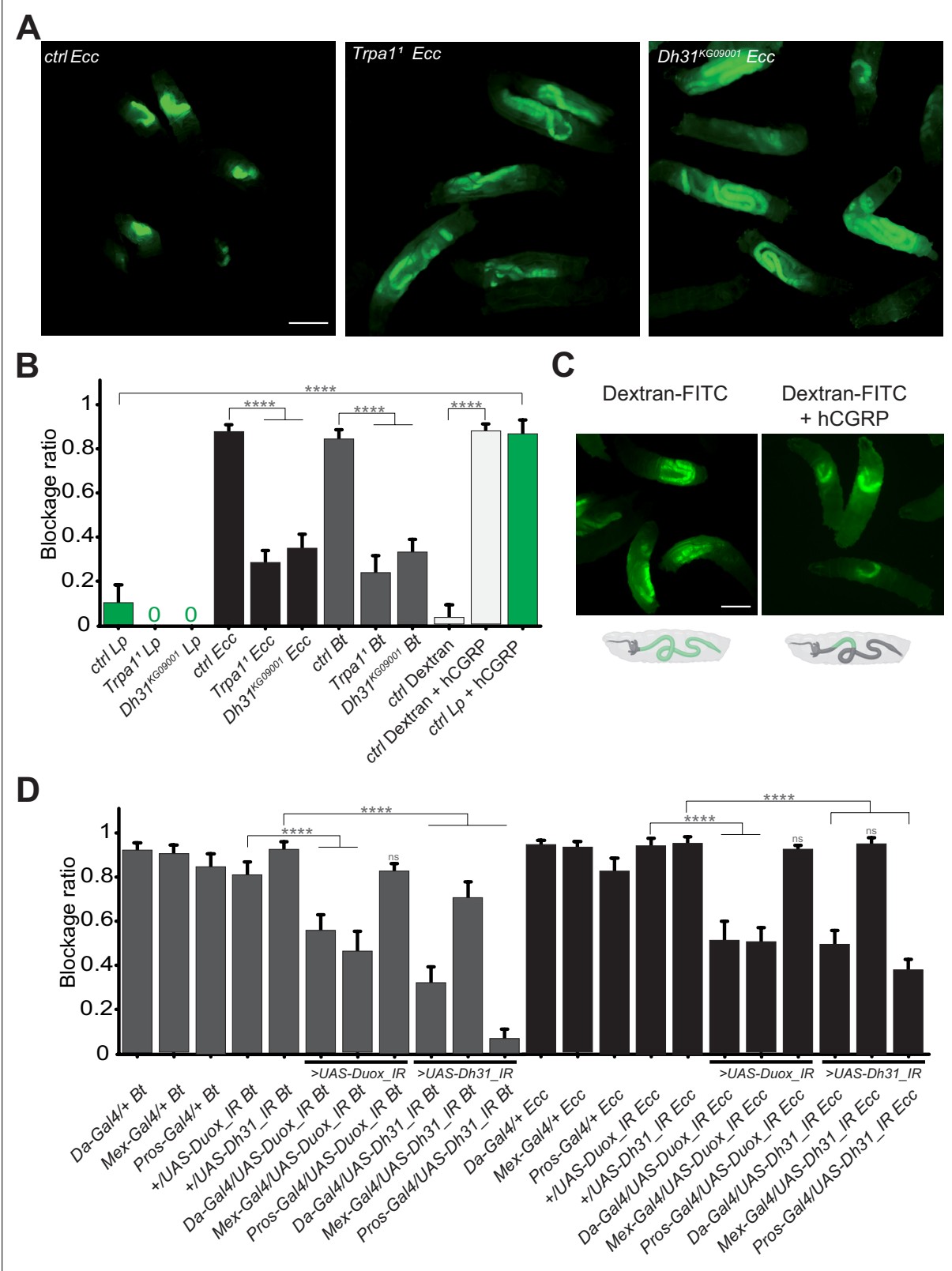

**Figure 3.** The bacterial blockage necessitates Duox in enterocytes, the TrpA1 channel and Dh31 in Pros + cells. (**A**) Pictures to illustrate the localization of the fluorescent bacteria within the intestine of control (ctrl), *TrpA1¹* or *Dh31^KG09001* L3 larvae after having been fed 1 hr with a mixture of yeast and *Ecc*. Scale bar is 1 mm. (**B**) Blockage ratio for control (ctrl) L3 larvae or mutants for *TrpA1¹* or *Dh31^KG09001* fed 1 hr with a mixture combining yeast and *Lp* or *Ecc* or *Bt* or fluorescent Dextran with or without hCGRP hormone. Shown is the average blockage ratio with a 95% confidence interval from at least three

*Figure 3 continued on next page*

*Figure 3 continued*

independent assays with at least 30 animals per condition and trial. 0 indicates an absence of blockage. **** indicates p<0.0001, Fisher exact t-test. See the source data file for details. (**C**) Pictures to illustrate the localization of the fluorescence within the intestine of control L3 larvae after having been fed 1 hr with a mixture of yeast and fluorescent Dextran with or without hCGRP hormone. Below the pictures are schematics representing larvae, their gut, and the relative position of the fluorescence in green. Scale bar is 1 mm. (**D**) Blockage ratio for animals expressing RNA interference constructions directed against *Duox* mRNA or *Dh31* mRNA, ubiquitously (Da-Gal4), in enterocytes (Mex-Gal4) or in enteroendocrine cells (Pros-Gal4) and then fed 1 hr with a mixture combining yeast and *Ecc* or *Bt*. Shown is the average blockage ratio with a 95% confidence interval from at least three independent assays with at least 30 animals per condition and trial. ns indicates values with differences not statistically significant, **** indicates p<0.0001, Fisher exact t-test. See the source data file for details.

The online version of this article includes the following figure supplement(s) for figure 3:

**Figure supplement 1.** *Ecc* is not blocked anteriorly in *TrpA1¹* and *Dh31ᴷᴳ⁰⁹⁰⁰¹* mutants, persists in the posterior part of the intestine and disappears while *Lp* can be blocked following exogenous addition of hCGRP.

**Figure supplement 2.** RNAi interference of *Dh31* in a specific subset of enteroendocrinal cells impairs the blockage phenotype.

phenomenon in Pros + cells (*Figure 3D*). In order to better delineate the cells required for the phenotype, we specifically silenced *Dh31* in a subset of Dh31 + EECs located at the anterior/acidic midgut junction using the DJ752-Gal4 line, which has been characterized in details in the study by *LaJeunesse et al., 2010*. Our results demonstrate that upon *Bt* or *Ecc* exposure, the parental DJ752-Gal4 line has a lower capacity to block the bacteria compared to the parental UAS control, but the combination with the *UAS-Dh31-IR* (DJ-Gal4/*UAS-Dh31_IR*) almost totally prevents the blockage phenomenon (*Figure 3—figure supplement 2*).

This data underscores the pivotal roles of Duox in ECs, Dh31 in Pros + cells and more specifically in the cells expressing the DJ752-Gal4 driver for the entrapment of pathogenic bacteria in the anterior part of the *Drosophila* larval midgut.

## Absence of ROS prevents blockage phenotype

Our data demonstrating that Duox protein is necessary for the confinement phenotype, implies that ROS generated by this enzyme are critical. The involvement of TrpA1, a known ROS sensor, further highlights the significance of these compounds in the process. To corroborate the necessity of ROS, we neutralized luminal ROS using DTT (Dithiothreitol), a potent reducing agent known for its efficacy against ROS, mixing it with the larval food (*Benguettat et al., 2018*). In normal conditions, larvae fed with *Bt* exhibit a compartmentalization of the bacteria in the anterior part of their intestine, as shown previously in *Figures 1A, B and 4A*. However, when the larvae were exposed to a mixture of *Bt* and DTT, the intestinal sequestration

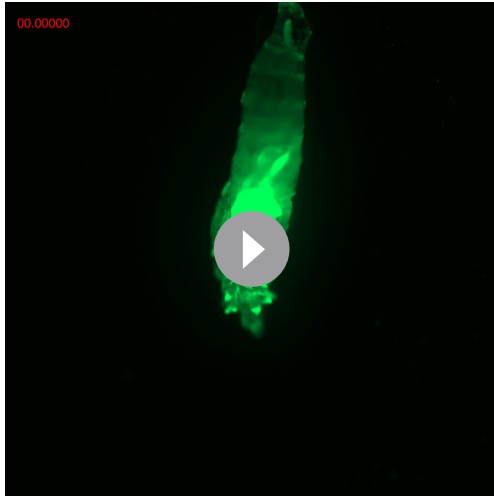

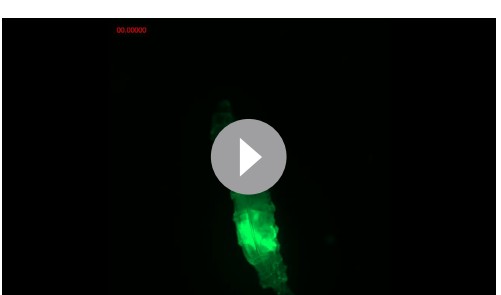

**Video 6.** Fluorescent *Ecc* is not blocked in the anterior part of the *TrpA1* mutant larval intestine and persists in the posterior midgut. Live imaging during 10 hr of a L3 *TrpA1* mutant larva previously fed 1 hr with a food containing *Ecc* fluorescent bacteria then transferred on a glass slide in a wet chamber. https://doi.org/10.6084/m9.figshare.25018463.v1.

https://elifesciences.org/articles/98716/figures#video6

**Video 7.** Fluorescent *Ecc* is not blocked in the anterior part of the *Dh31* mutant larval intestine and persists in the posterior midgut. Live imaging during 10 hr of a L3 *Dh31* mutant larva previously fed 1 hr with a food containing *Ecc* fluorescent bacteria then transferred on a glass slide in a wet chamber. https://doi.org/10.6084/m9.figshare.25018472.v1.

https://elifesciences.org/articles/98716/figures#video7

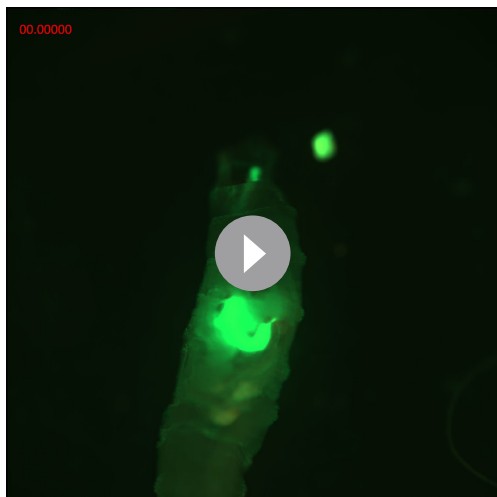

**Video 8.** Fluorescent *Lp* is blocked in the anterior part of the larval intestine following treatment with hCGRP. Live imaging during 12 hr of a control L3 larva previously fed 1 hr with a food containing *Lp* fluorescent bacteria and hCGRP then transferred on a glass slide in a wet chamber. https://doi.org/10.6084/m9.figshare.25018481.v1.

https://elifesciences.org/articles/98716/figures#video8

of bacteria in the anterior midgut was abolished (*Figure 4A*). The effect of DTT over the blockage phenotype was quantified using DTT mixed to Dextran-FITC in order to compare with a condition leading to an almost full posterior localization (*Figure 4B*).

Evidence that host ROS were the elements that initiated the blocking mechanism raised the question of discrimination between bacteria. Indeed, there are studies linking the chemical signature of pathogens, particularly Uracil, with the release of ROS in the intestinal lumen of adult *Drosophila* (*Kim et al., 2020*; *Lee et al., 2013*). *Ecc* and *Bt* are precisely bacteria that release Uracil, unlike *Lp* (*Lee et al., 2013*). To determine whether it was possible to induce *Lp* blockade by adding Uracil, we used our intoxication protocol supplemented with Uracil. Using an Uracil concentration that was previously demonstrated as sufficient to induce ROS production in adults (*Lee et al., 2013*), we did not observe the blockage of *Lp* in the anterior part of the intestine (*Figure 4—figure supplement 1*).

Together, our results strongly suggest a discrimination in the larval midgut lumen between commensal and pathogenic bacteria. Such a discrimination involves the Duox enzyme in producing ROS, and these ROS act as key signals initiating the confinement mechanism. ROS may be produced first to directly limit bacterial proliferation but meanwhile trigger a cascade of events leading to sequestration in the anterior part.

## Blockage is crucial for bacterial elimination and larval survival

Our real-time observations showed that between 4 and 6 hr after the bacterial compartmentalization of *Bt* or *Ecc* in the anterior midgut, the bacteria disappeared (*Videos 1 and 2*, *Figure 1—figure supplement 1A and B*). This led us to investigate the relationship between pathogen localization and larval survival. The hypothesis was that the disappearance of the GFP signal corresponded to the bacterial death. We therefore assessed the *Lp*, *Bt,* or *Ecc* load over time in dissected midguts of control larvae previously exposed to contaminated food and then transferred to a wet chamber without food. Consistent with our film data, while the *Lp* load remained stable, the quantities of *Ecc* and *Bt* diminished rapidly in the intestines of control larvae by 4 hr, with no bacteria detectable 8 hr post-ingestion and blockage (*Figure 5A, B and C*). However, in the intestines of *TrpA1[1]* and *Dh31[KG09001]* mutants, and despite a global intestinal immune response comparable to what was measured in control animals (*Figure 5—figure supplement 1*), the amount of *Bt* and *Ecc* increased overtime (*Figure 5B and C*). Interestingly, the quantity of bacteria presents in the intestine after a 1 hr feeding period was consistent across different host genetic backgrounds for any given bacterial species. This uniformity highlights the process' active regulation rather than attributing the observed patterns to a simple cessation of food intake or a global disruption of peristalsis in *Dh31* and *TrpA1* mutants resulting in a massive and uncontrolled influx of bacteria into the posterior part of the intestine. To verify that the mutants we were using were capable of intestinal muscle movements, we filmed the guts of the *Dh31[KG09001]* mutant larvae. We observed strong waves of contractions in the intestines of the *Dh31[KG09001]* mutant animals, whether they were fed with Ecc (*Video 9*) or not (*Video 10*). Unfortunately, we were unable to assess the bacterial load in the tested mutants beyond 6 hr due to the deterioration of the midguts. Supporting this, in *Videos 6 and 7*; *Figure 3—figure supplement 1A and B*, we observed that *Bt* or *Ecc* bacteria which were not blocked in the anterior part of the midgut did not disappear over time. More importantly, *TrpA1[1]* and *Dh31[KG09001]* mutant larvae containing *Bt*

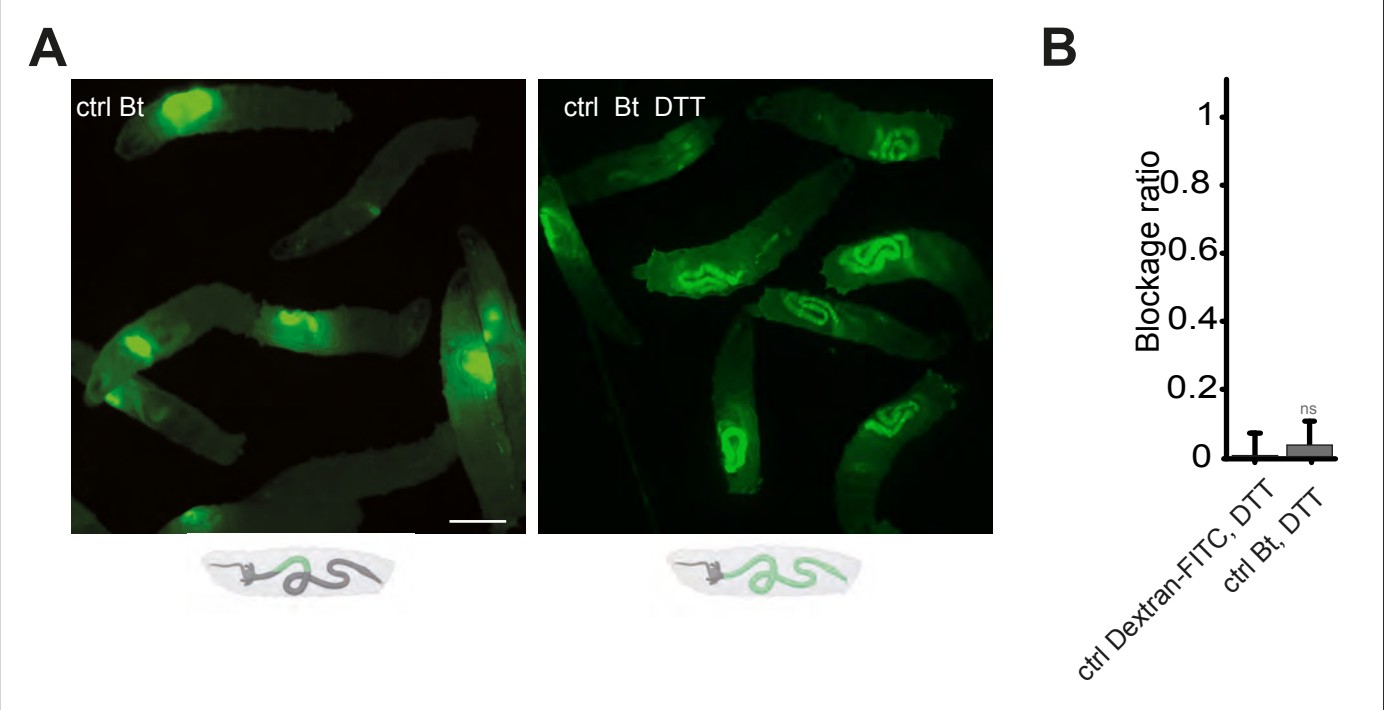

**Figure 4.** Blocking the ROS with DTT prevents the compartmentalization of *Bt,* and the larvae with bacteria in the posterior part of the intestine die. (**A**) Pictures to illustrate the localization of the fluorescence within the intestine of control (ctrl) L3 larvae after having been fed 1 hr with a mixture of yeast and *Bt* with or without DTT. Below the pictures are schematics representing larvae, their gut, and the relative position of the fluorescent bacteria in green. Scale bar is 1 mm. (**B**) Blockage ratio for control (ctrl) L3 larvae fed 1 hr with a mixture combining yeast, DTT and fluorescent Dextran or *Bt*. Shown is the average blockage ratio with a 95% confidence interval from at least three independent assays with at least 18 animals per condition and trial. ns indicates values with differences not statistically significant, Fisher exact t-test. See the source data file for details.

The online version of this article includes the following figure supplement(s) for figure 4:

**Figure supplement 1.** Uracil supplementation does not trigger the blockage phenotype.

or *Ecc* stopped to move suggesting they were dead. We confirmed the precocious death of the *TrpA1[1]* and *Dh31[KG09001]* larvae fed with either *Bt* or *Ecc* compared to control animals (***Figure 5D and E***). Importantly, these mutants exhibited sustained viability overnight in the wet chamber after having been fed 1 hr with a mixture containing *Lp* or a bacteria-free diet (***Figure 5E***). Interestingly, control larvae fed a mixture of *Bt* and DTT, which neutralizes ROS and thus inhibits the confinement, also perished (***Figure 5F***). This mortality was not due to DTT, as larvae fed Dextran-FITC plus DTT survived (***Figure 5F***). These findings suggest that, in larvae, the blockage of *Bt* and *Ecc* in the anterior part of the midgut – involving a sequence of events with ROS production by Duox, TrpA1 activation by ROS, and Dh31 secretion – is essential for bacterial elimination by the host. Thus, failure to compartmentalize pathogenic bacteria like *Ecc* or *Bt* results in their proliferation and consequent larval death.

## The confined area is delimited by TrpA1+/Dh31+ cells and muscular structures

The above results suggest a working model involving the ROS/TrpA1/Dh31 axis in which Dh31 release from EECs leads to muscle contractions. However, unlike in adult *Drosophila*, where bacteria are expelled from the gut, in larvae, we observed a blockage mechanism. To better understand the physiology of the process, we utilized confocal microscopy to thoroughly examine larval midguts and explore the relationship between TrpA1-positive (TrpA1+) cells, anterior confinement of the bacteria, and muscular structures. In larval midguts, TrpA1 + cells were also Dh31+, and these Dh31 + cells were identified as EECs (Pros+), typically located at the end of the anterior midgut. With the reporter line we used (*TrpA1*-Gal4/UAS-RFP), we noted an average of 3 TrpA1+/Dh31 + cells per gut (ranging from 2 to 6 cells across 14 examined guts, see source data file; ***Figure 6A–F***). Interestingly, these cells may be the ones described previously as being part of a valve (***LaJeunesse et al., 2010***) and also

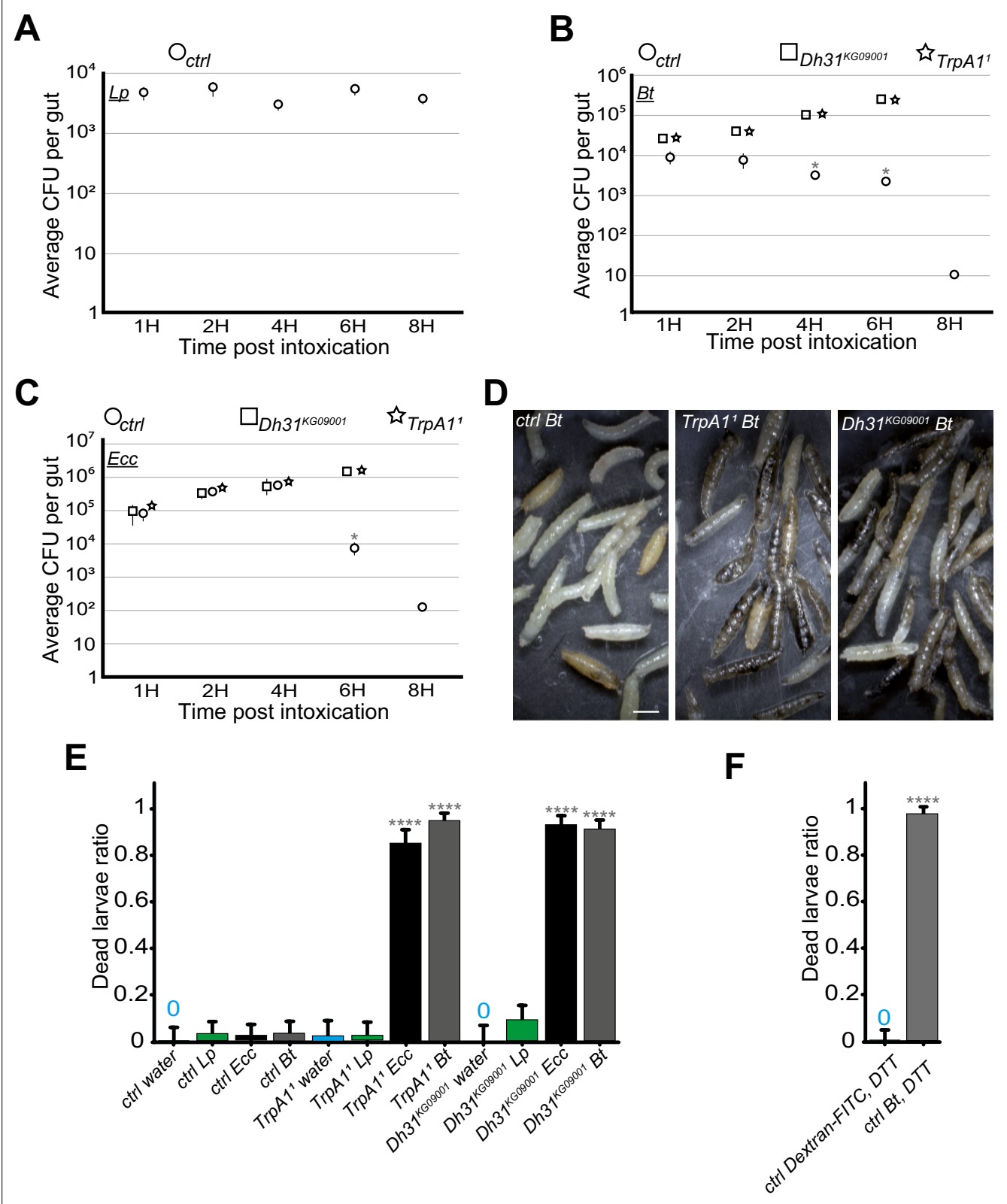

**Figure 5.** In the absence of blockage in *TrpA1¹* or *Dh31^KG09001^* mutants, *Bt* and *Ecc* proliferate in the larval intestine and the larvae die. (**A, B and C**) quantification over time of the amount of *Lp*, (**A**), *Bt* (**B**) or *Ecc* (**C**) live bacteria within the larval intestine of control (ctrl) (**A, B and C**), *Dh31^KG09001^* (**B and C**) and *TrpA1¹* (**B and C**) animals following a 1 hr feeding period with a solution containing yeast and bacteria. CFU stands for Colony Forming Units. Shown is the average ± SEM of at least three independent experiments with at least 7 guts each. After 8 hr, either all the *TrpA1¹* or *Dh31^KG09001^* larvae

*Figure 5 continued on next page*

*Figure 5 continued*

were dead or the intestines were severely damaged preventing the CFU counting. * Indicates p<0.05, Mann Whitney, two-tailed test. See the source data file for details. (**D**) Pictures of control (ctrl) or *TrpA1¹* or *Dh31^{KG09001}* larvae after 8 hr in a wet chamber following a 1 hr feeding with a mixture of yeast and *Bt*. For control larvae, some animals made pupae that are visible while for *TrpA1¹* and *Dh31^{KG09001}* mutants, the dark larvae are dead non-moving melanized animals. Scale bar is 1 mm. (**E**) Ratio of dead control or *TrpA1¹* or *Dh31^{KG09001}* larvae after 8 hr in a wet chamber following or not (water) a 1 hr feeding period with yeast mixed with *Lp* or *Ecc* or *Bt*. Shown is the average with 95% confidence interval of at least three independent experiments with at least 21 larvae per trial and condition. The 0 symbol indicates an absence of lethality. **** indicates p<0.0001, Fisher exact t-test. See the source data file for details. (**F**) Ratio of dead control (ctrl) larvae after 8 hr in a wet chamber following a 1 hr feeding period with a mixture combining yeast, DTT and Dextran fluorescent beads or *Bt*. Shown is the average with 95% confidence interval of at least three independent experiments with at least 18 larvae per trial and condition. The 0 symbol indicates an absence of lethality. **** indicates p<0.0001, Fisher exact t-test. See the source data file for details.

The online version of this article includes the following figure supplement(s) for figure 5:

**Figure supplement 1.** The intestinal immune response of reference animals, *TrpA1¹* and *Dh31^{KG09001}* mutants are similar.

reported by Zaidman-Rémy and colleagues as being Dh31 + as well as Hml+, an hemocyte marker (*Zaidman-Rémy et al., 2012*). The identity of these cells was in agreement with our genetic and functional data linking Dh31 with Pros + cells including EECs (*Figure 3D*) and suggested that TrpA1 and Dh31 operate within the same cells (*Figure 6C, C', and C''*). The shape of these TrpA1 +Dh31 + cells was characteristic of EECs (*Figure 6A'*). In agreement with a model involving an interaction of secreted ROS with TrpA1 and a subsequent local Dh31 release to act on muscles, following exposure to food contaminated with fluorescent *Bt* or *Ecc*, the bacteria were confined in an area delimited by the anterior part of the gut and the TrpA1 + cells patch (*Figure 6B*). Additionally, we observed that the amount of Dh31 within Pros + cells of larvae confining bacteria was lower compared to those not exhibiting the blockage, such as *TrpA1* mutant (*Figure 6D and E*). Then, we wondered whether specific muscle structures would exist close to the TrpA1 + cells in the anterior midgut. Actin labeling revealed fibrous structures on the basal side of the gut and attached to it in a transversal position (*Figure 6C'', and F–H'*). These structures, typically lost during dissection, have been described previously, and the hypothesis of their connection with a valve at the junction with the midgut was proposed in a report studying larval midgut peristalsis (*LaJeunesse et al., 2010*). Notably, the attachment points of these filaments, or tethers, corresponded with the locations of TrpA1 + cells and the boundary of the area where *Ecc* or *Bt* were confined (*Figure 6C'', and F–H*). These filaments have been described as longitudinal muscles emanating from two out of the four gastric caeca, but this might be a misinterpretation of the images generated by *LaJeunesse et al., 2010*. Indeed, a recent study describes these muscular structures using in vivo observations without dissections. Interestingly, they show that these muscles belong to a subgroup of alary muscles named TARMs (thoracic alary related muscles;

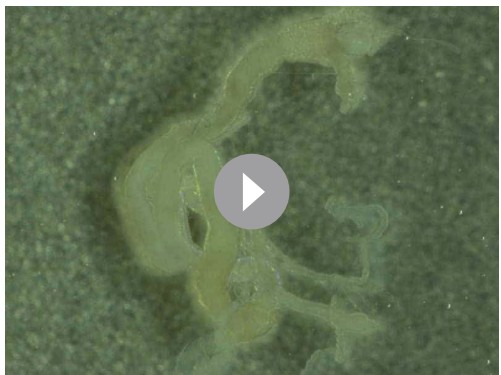

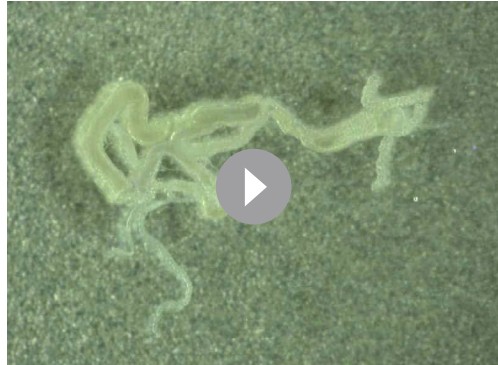

**Video 9.** Robust muscular contraction waves are observed in intestines of *Dh31* mutant larvae previously fed with *Ecc*. Live imaging of a L3 *Dh31* mutant larval intestine. The animal was previously fed 1 hr with a food containing *Ecc* fluorescent bacteria and the dissected gut was then transferred in Schneider media. https://doi.org/10.6084/m9.figshare.27100474.v1. https://elifesciences.org/articles/98716/figures#video9

**Video 10.** Robust muscular contraction waves are observed in intestines of *Dh31* mutant larvae. Live imaging of a L3 *Dh31* mutant larval intestine. The animal was previously fed 1 hr with a food mixture that did not contain bacteria and the dissected gut was then transferred in Schneider media. https://doi.org/10.6084/m9.figshare.27100483.v1. https://elifesciences.org/articles/98716/figures#video10

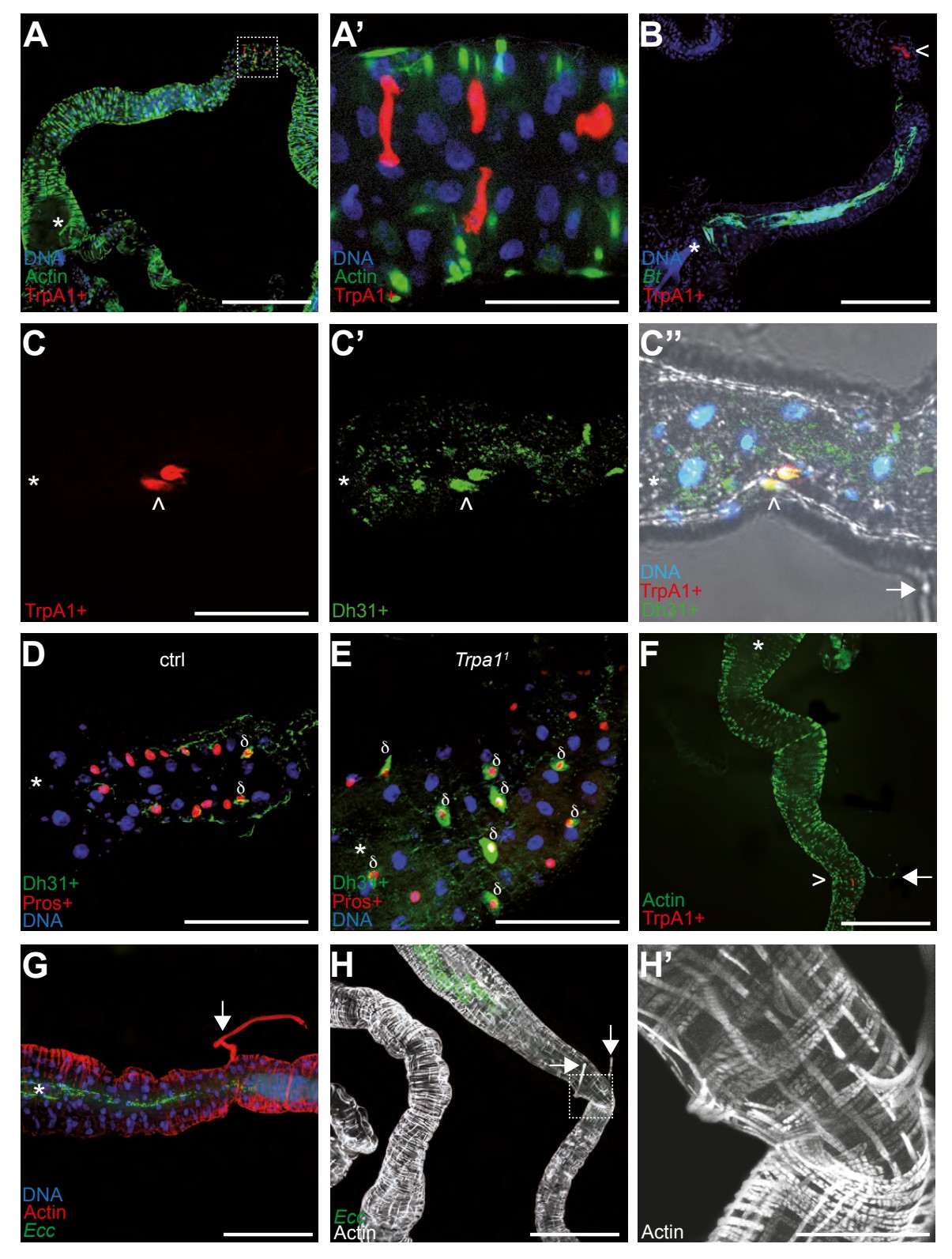

**Figure 6.** TrpA1 + cells in the gut are enteroendocrine cells concentrated in a portion of the intestine bordering the blocked bacteria. Confocal fluorescent pictures of the anterior portions of L3 larval intestines to detect; longitudinal and transversal muscles concentrated in actin (**F, G, H and H'**), TrpA1 + cells producing RFP (**A, A', B, C, C'' and F**), GFP-bacteria (**B, G and H**), Dh31 + cells (**C', C'', D and E**), Pros + cells (**D and E**) and nuclei with DNA staining (**A, A', B, C'', D, E and G**). In B, D, E, G, H, and H'; animals were previously fed for 1 hr with a mixture containing bacteria and yeast

*Figure 6 continued on next page*

*Figure 6 continued*

with *Bt* (**B, D and E**) or *Ecc* (**G, H and H'**). When present, the white star indicates the anterior part of the intestinal portion shown, the arrows point to TARMs and the > symbols point to TrpA1 + cells. The empty squares in A and H with dashed lines correspond to the portion of the image magnified in A' and H', respectively. Scale bar in A, B, F, G, and H represents 500 μm, in A', C, D, E, and H' represents 100 μm.

The online version of this article includes the following figure supplement(s) for figure 6:

**Figure supplement 1.** TARMsT2 are attached to the longitudinal gut muscles.

**Figure supplement 2.** TARMs T2 structures are still present in *TrpA1¹* and *Dh31^KG09001* mutant backgrounds.

*Bataillé et al., 2020*). Specifically, TARMsT1 connect the anterior of the larvae to the extremities of gastric caeca, while TARMsT2 link the anterior part of the gut to the larval epidermis. Our findings support the hypothesis that the observed muscular structures close to the TrpA1 + cells are TARMsT2 (*Figure 6C" and F–H'*, *Figure 6—figure supplement 1* and *Video 11*). These TARMsT2 are attached to the longitudinal gut muscles and the intestine forms a loop at the attachment site (*Figure 6H'* and *Figure 6—figure supplement 1*; *Bataillé et al., 2020*). The presence of Dh31 + EECs in this specific curved region of the gut close to the TARMsT2 attachment (*Figure 6C''*) led to the hypothesis that this region may act like a valve (*LaJeunesse et al., 2010*). Importantly, we confirmed that these TARMsT2 are still present in *TrpA1¹* and *Dh31^KG09001* larvae (*Figure 6—figure supplement 2*). Our genetic, functional and physiological data confirm the model describing the valve and reveal a new and crucial role in the context of pathogenic bacteria ingestion.

## IMD pathway is mandatory for eliminating trapped bacteria

In our study, we observed that control larvae were able to kill *Ecc* and *Bt* bacteria trapped in the anterior part of the gut within 6–8 hr (*Videos 1 and 2* and *Figure 5A–C*). This finding raised questions about the mechanism of bacterial elimination and the potential role of the IMD pathway in this process. While larval intestinal immunity is multifaceted, a key defense mechanism against bacteria is the production and secretion of AMPs (*Hanson and Lemaitre, 2020*). Thus, we focused our investigations on AMPs. Both *Ecc* and *Bt* possess DAP-type peptidoglycans (PGN), known to activate the IMD signaling cascade, which leads to the production of AMPs like Diptericin (*Kaneko et al., 2006*; *Leulier et al., 2003*; *Stenbak et al., 2004*). We used various mutants deficient in components of the IMD pathway, including PGRP-LC and PGRP-LE (PGN receptors), Dredd (an intracellular component), and Relish (a NF-kB transcription factor; *Zhai et al., 2018*). Additionally, we studied a mutant, *ΔAMP14*, lacking 14 different AMPs (*Carboni et al., 2022*). We first assayed whether the IMD pathway was required for the blockage phenotype upon ingestion of *Bt* or *Ecc*. While *Lp* was distributed throughout the gut of IMD pathway mutants, *Ecc* and *Bt* were confined to the anterior part of the intestine, akin

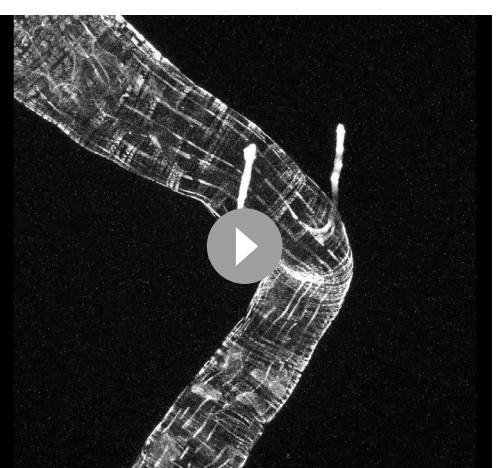

**Video 11.** TARMsT2 are attached to the longitudinal gut muscles. Confocal imaging of the intestine from a control animal stained with fluorescent phalloidin and animated 3D-reconstruction of the anterior portion containing the attached TARMs. https://doi.org/10.6084/m9.figshare.25018496.v1.

https://elifesciences.org/articles/98716/figures#video11

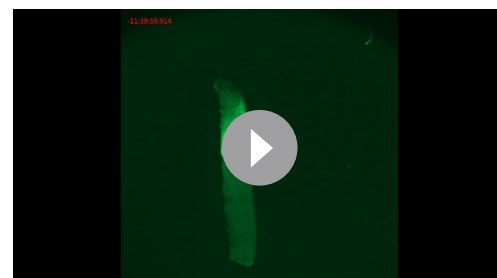

**Video 12.** Fluorescent *Bt* is blocked in the anterior part of the *PGRP-LC* mutant larval intestine and persists. Live imaging during 12 hr of a L3 *PGRP-LC* mutant larva previously fed 1 hr with a food containing *Bt* fluorescent bacteria then transferred on a glass slide in a wet chamber. https://doi.org/10.6084/m9.figshare.25018499.v1.

https://elifesciences.org/articles/98716/figures#video12

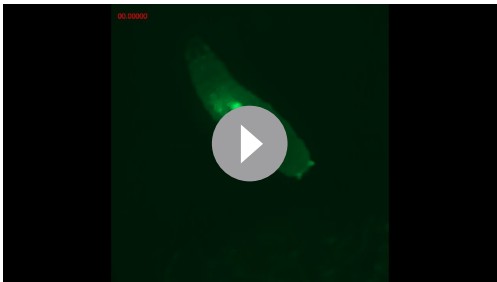

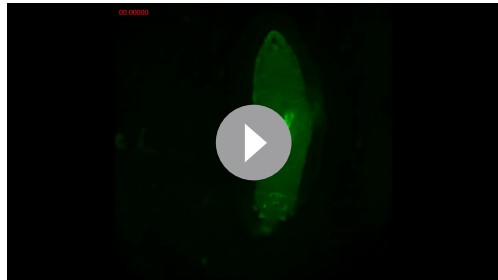

**Video 13.** Fluorescent *Ecc* is blocked in the anterior part of the *PGRP-LE* mutant larval intestine and persists. Live imaging during 12 hr of a L3 *PGRP-LE* mutant larva previously fed 1 hr with a food containing *Ecc* fluorescent bacteria then transferred on a glass slide in a wet chamber. https://doi.org/10.6084/m9.figshare.25018505.v1.

https://elifesciences.org/articles/98716/figures#video13

**Video 15.** Fluorescent *Bt* is blocked in the anterior part of the *Rel* mutant larval intestine and persists. Live imaging during 12 hr of a L3 *Rel* mutant larva previously fed 1 hr with a food containing *Bt* fluorescent bacteria then transferred on a glass slide in a wet chamber. https://doi.org/10.6084/m9.figshare.25018529.v1.OI: 10.

https://elifesciences.org/articles/98716/figures#video15

to control larvae (*Videos 12–16*, *Figure 7A and B* and *Figure 7—figure supplements 1 and 2*). Thus, the IMD pathway is not required for the compartmentalization of *Ecc* and *Bt* in larval intestines. Nevertheless, the movies suggested a death of the IMD mutant larvae despite the blockage of either *Bt* or *Ecc*. We therefore tested the survival of these IMD pathway mutants following a 1 hr feeding with a mixture containing or not (water) fluorescent bacteria followed by a transfer into a humid chamber. While neither control animals nor the IMD pathway mutants died following a 1 hr feeding period with a *Lp* contaminated or non-contaminated food and transfer into a humid chamber, all the IMD pathway mutants, including *ΔAMP14*, had a decreased survival after exposure to *Ecc* or *Bt* (*Figure 7C and D*). Thus, the IMD pathway is central for the survival of these animals with bacteria blocked in the anterior part of the intestine. As this increased lethality in IMD pathway mutants might be related to an uncontrolled growth of the confined *Bt* and *Ecc* bacteria, we performed CFU counting. With *Bt* and *Ecc*, while the initial inoculum was divided by $10^3$ in 8 hr in the control larvae, the bacterial population was maintained and even increased 10-fold in IMD pathway mutants including *ΔAMP14* (*Figure 7E and F*). Additional observations from filming the fate of *Bt* and *Ecc* in IMD pathway mutant larvae confirmed these findings (*Videos 12–16*). The GFP-bacteria, although sequestered in the anterior part of the intestine, did not disappear, coinciding with larval immobility and presumed death. In order to delineate whether the localization of the bacteria within the intestine coincides with the gut area producing the AMPs, we assayed the spatial and temporal activation of the AMP Diptericin-encoding gene (Dpt) using a reporter line intoxicated with fluorescent bacteria. We observed that larvae fed with *Lp* do not robustly express the reporter gene

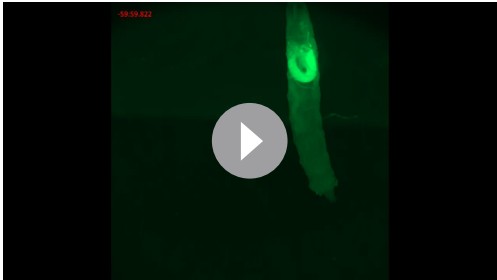

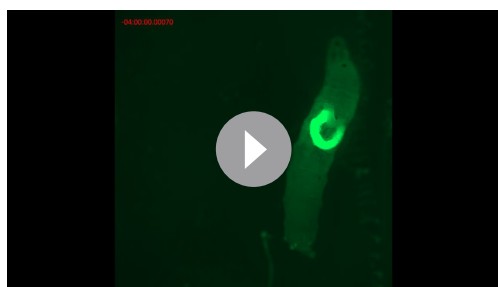

**Video 14.** Fluorescent *Ecc* is blocked in the anterior part of the *Dredd* mutant larval intestine and persists. Live imaging during 12 hr of a L3 *Dredd* mutant larva previously fed 1 hr with a food containing *Ecc* fluorescent bacteria then transferred on a glass slide in a wet chamber. https://doi.org/10.6084/m9.figshare.25018517.v1.

https://elifesciences.org/articles/98716/figures#video14

**Video 16.** Fluorescent *Ecc* is blocked in the anterior part of the *Rel* mutant larval intestine and persists. Live imaging during 12 hr of a L3 *Rel* mutant larva previously fed 1 hr with a food containing *Ecc* fluorescent bacteria then transferred on a glass slide in a wet chamber. https://doi.org/10.6084/m9.figshare.25018538.v1.

https://elifesciences.org/articles/98716/figures#video16

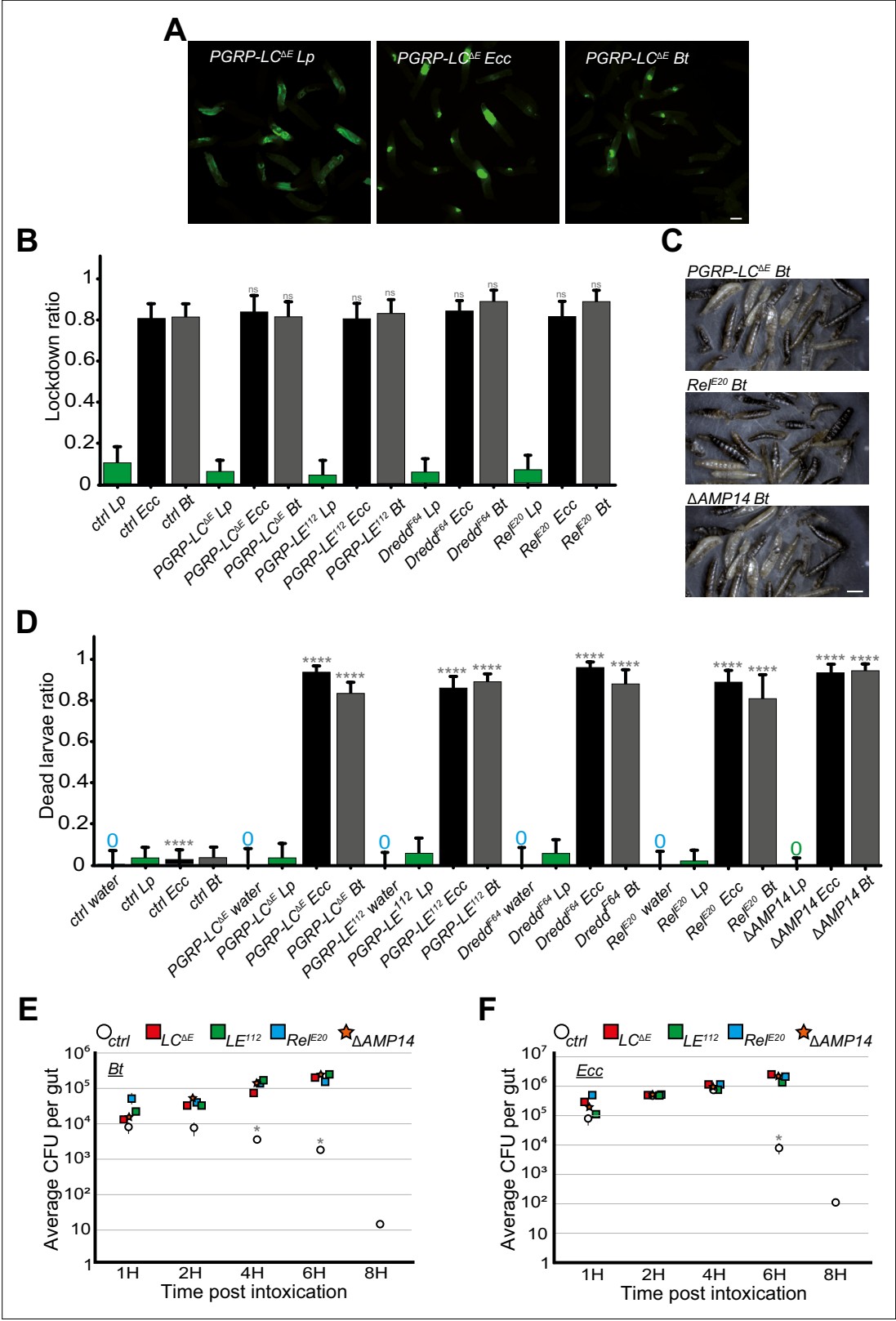

**Figure 7.** IMD pathway is not required for the blockage but essential for larvae survival and *Bt* or *Ecc* clearance. (**A**) Pictures to illustrate the localization of the fluorescence within the intestine of *PGRP-LC^{ΔE}* L3 larvae after having been fed 1 hr with a mixture of *Lp* or *Ecc* or *Bt*. Scale bar is 1 mm. (**B**) Blockage ratio for control L3 larvae or mutants of the IMD pathway fed 1 hr with a mixture combining yeast and *Lp* or *Ecc* or *Bt*. Shown is the average with 95% confidence interval of at least three independent experiments with at least 20 larvae per trial and condition. ns indicates values with differences

*Figure 7 continued on next page*

*Figure 7 continued*

not statistically significant, Fisher exact t-test. See the source data file for details. (**C**) Pictures of *PGRP-LC^ΔE^*, *Rel^E20^* or *ΔAMP14* mutant larvae after 18 hr in a wet chamber following a 1 hr feeding with a mixture of yeast and *Bt*. The dark larvae are dead non-moving melanized animals. *ΔAMP14* is a mutant deleted for 14 antimicrobial-encoding genes. (**D**) Ratio of dead control or *TrpA1^1^* or *Dh31^KG09001^* larvae after 18 hr in a wet chamber following or not (water) a 1 hr feeding period with yeast mixed with *Lp* or *Ecc* or *Bt*. Shown is the average with 95% confidence interval of at least three independent experiments with at least 20 larvae per trial and condition. The 0 symbol indicates an absence of lethality. **** indicates p<0.0001, Fisher exact t-test. See the source data file for details. (**E and F**) quantification over time of the amount of *Bt* (**A**) and *Ecc* (**B**) live bacteria within the larval intestine of control or IMD pathway mutant animals including *ΔAMP14* following a 1 hr feeding period with a solution containing yeast and bacteria. CFU stands for Colony Forming Units. *ΔAMP14* is a mutant deleted for 14 antimicrobial-encoding genes. Shown is the average ± SEM of at least three independent experiments with at least seven guts each. After 8 hr, either all the mutants were dead or the intestines were severely damaged preventing the CFU counting. * Indicates p<0.05, Mann Whitney, two-tailed test. See the source data file for details.

The online version of this article includes the following figure supplement(s) for figure 7:

**Figure supplement 1.** *Bt* and *Ecc* are blocked anteriorly and persist in *PGRP-LC^ΔE^* and *PGRP-LE^112^* mutants, respectively.

**Figure supplement 2.** *Bt* and *Ecc* are blocked and persist anteriorly in *Dredd^F64^* and *Rel^E20^* mutants.

**Figure supplement 3.** The *Diptericin* gene is expressed in the anterior part of the gut following the blockage.

in the posterior part of the intestine, while the bacteria can be seen in this area. Conversely, transgenic animals infected with *Bt* and blocking the bacteria in the anterior part of the gut robustly induce the expression of the AMP-reporter gene in this area (*Figure 7—figure supplement 3*). In conclusion, our findings illustrate that although the IMD pathway is dispensable for the initial compartmentalization of pathogenic bacteria, a process contingent on the ROS/TrpA1/Dh31 axis, it plays a crucial role in their subsequent elimination. Indeed, the AMPs produced following IMD pathway activation are essential for killing the trapped bacteria and ensuring larval survival (*Figure 8*).

## Discussion

Leveraging the transparency of the *Drosophila* larvae, we have successfully developed a novel real-time experimental system to monitor the fate of fluorescent bacteria ingested along with food. This methodological advancement has enabled us to unveil a previously uncharacterized physiological pathway necessary for the efficacy of the larval intestinal immune response. Our research has uncovered a unique defense mechanism centered around the presumed valve located in the anterior midgut, regulated by the enteroendocrine peptide Dh31 (*LaJeunesse et al., 2010*). Notably, we observed that the pathogenic bacteria we tested, were confined to the anterior section of the larval intestine as early as 15 min post-ingestion. We determined that this intestinal sequestration of pathogenic bacteria necessitates a ROS/TrpA1/Dh31 axis initiated by Duox activity in ECs in response to pathogenic bacteria. We suspect the secreted ROS to interact with the TrpA1 ion channel receptor located in Dh31-expressing EECs adjacent to the valve-like structure (*Figure 8*). Previous studies on the interaction between ROS and TrpA1 support our hypothesis (*Ogawa et al., 2016*). The confining of pathogenic bacteria to the anterior part of the larval intestine is a mandatory step prior to their subsequent elimination by the IMD pathway. Intriguingly, previous studies utilizing fluorescent bacteria have already highlighted a specific localization of pathogenic bacteria in the larval gut. Bacteria such as *Ecc15*, *Pseudomonas entomophila*, *Yersinia pestis*, *Salmonella enterica serovar Typhimurium*, and *Shigella flexneri* were observed predominantly in the anterior part of the larval gut 6 hr after oral infection (*Basset et al., 2000*; *Bosco-Drayon et al., 2012*; *Earl et al., 2015*; *Ramond et al., 2021*; *Vodovar et al., 2005*). Our findings suggest that these bacteria can all induce ROS production by ECs, providing a unifying mechanism for their containment and elimination in the larval gut. Interestingly, it has been reported that the opportunistic pathogen *Staphylococcus aureus (strain USA300)* predominantly colonizes the posterior midgut of *Drosophila* larvae, leading to the death of 93% of the larvae (*Ramond et al., 2021*). This strain of *S. aureus* produces high levels of detoxifying enzymes, such as catalase and superoxide dismutases, which effectively neutralize ROS. The authors suggested that the neutralization of ROS bactericidal activity by these enzymes is directly responsible for the bacterial proliferation and consequent host mortality (*Ramond et al., 2021*). However, considering our findings, we propose a complementary or an alternative interpretation: the neutralization of ROS by these detoxifying enzymes might prevent the bacterial compartmentalization, thereby allowing *S. aureus* to access and establish in the posterior midgut. Our data also indicate that when pathogenic

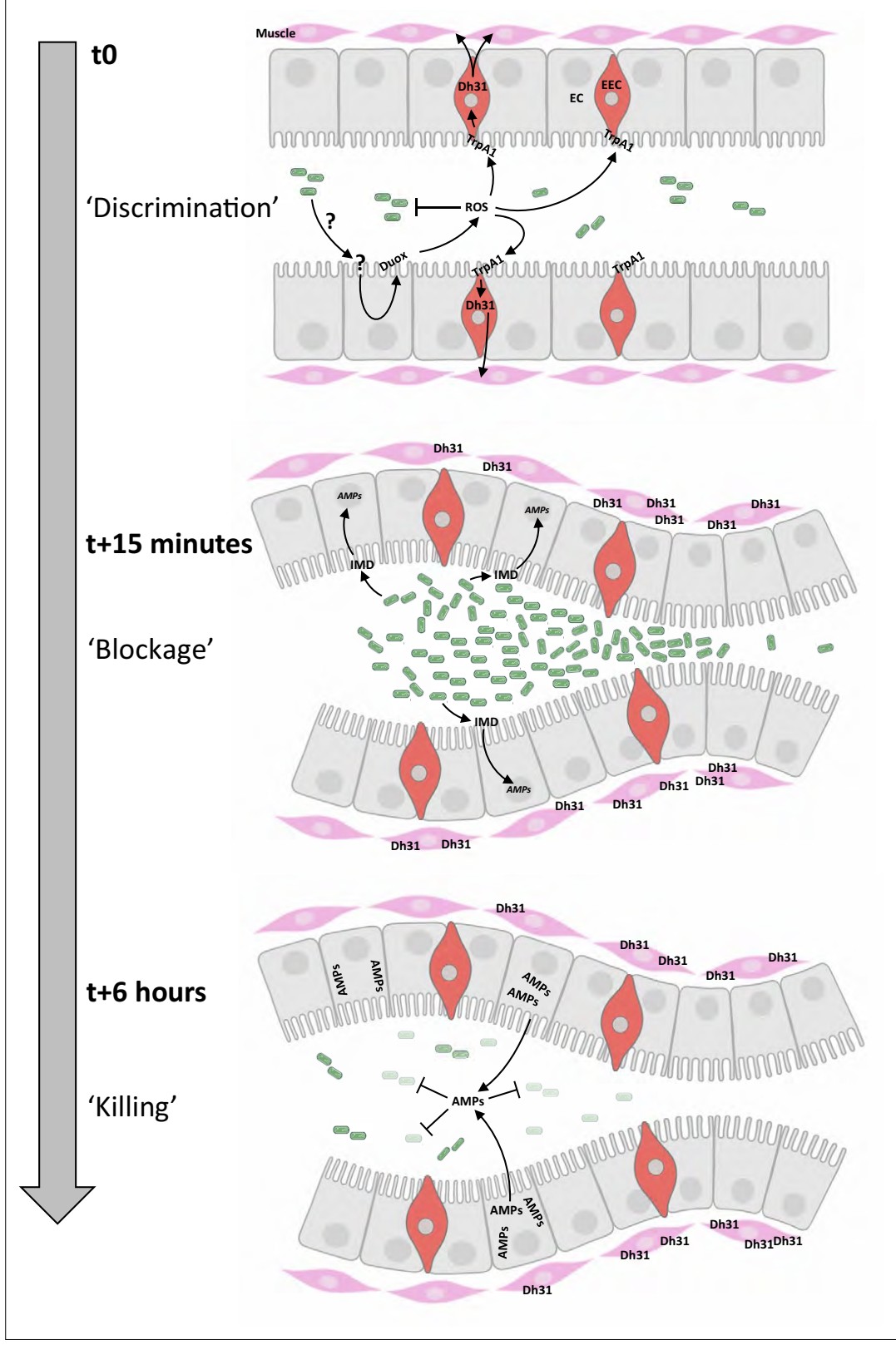

**Figure 8.** Chronological coordination of ROS/TrpA1/Dh31 and IMD pathways for an efficient microbial elimination. t0: larvae ingest bacteria from the food mixture (anterior on the left, only bacteria similar to *Ecc* or *Bt* are illustrated). This initial phase necessitates a discrimination between commensal and pathogenic bacteria, not elucidated in this study (symbolized by '?'). The presence of pathogenic bacteria induces the production of ROS

*Figure 8 continued on next page*

*Figure 8 continued*

by enterocytes (EC) in a Duox-dependent manner. Then ROS activates TrpA1 in enteroendocrine cells (EEC). t15 minutes: Dh31 secretion by EEC is responsible for the blockage of bacteria likely by promoting visceral muscle contractions leading to a closure of a valve-like structure. This phenomenon concentrates the bacteria in the anterior part of the gut. The bacterial concentration in this part of the intestinal lumen may facilitate the triggering of the IMD signaling cascade that controls the transcription of the genes (*AMPs*) encoding the antimicrobial peptides (AMPs). t6 hours: the valve-like structure is still closed. The bactericidal activity of AMPs has eliminated most of the bacteria accumulated in the anterior part of the intestine. Importantly, if confinement is prevented, the larvae die; if the response by antimicrobial peptides is hindered, the larvae die.

bacteria reach the posterior midgut, as observed in larvae fed with DTT or in *TrpA1* or *Dh31* mutants, larval survival is significantly jeopardized. This suggests that larval mortality could be attributed not to the inhibition of ROS in the posterior midgut but rather to the presence of bacteria in this region. In the study by *Ramond et al., 2021* involving *S. aureus*, the observation of bacterial spread into the posterior part was recorded 3 hr post-infection. It would be insightful to further investigate the dynamics of bacterial diffusion to determine whether, like other pathogens we studied, *S. aureus* is initially confined to the anterior part of the gut shortly after ingestion, specifically around 15 min post-infection. This could provide a broader understanding of the interplay between pathogen-specific strategies and host defense mechanisms in *Drosophila* larvae.

Our study has focused on an elbow-shaped region in the *Drosophila* larval midgut, characterized by a narrowing of the lumen and surrounded by muscular fibers. This region, where we observed a halt in transit when food is contaminated, has been aptly termed a valve due to its functional similarity to the human pyloric region. Interestingly, *LaJeunesse et al., 2010* reported the presence of muscular fibers in this area while *Bataillé et al., 2020* identified them as a subgroup of alary muscles known as TARMs. Specifically, TARMsT1 connect the anterior part of the larvae to the extremities of a pair of gastric caeca, whereas TARMsT2 link the anterior part of the intestine to the larval epidermis. Our actin staining corroborates the identification of these muscles as TARMsT2, which are attached to the longitudinal muscles of the intestine, causing the intestine to form a loop at the site of attachment (*Bataillé et al., 2020*). The presence of Dh31-positive EECs in this specific elbow-shaped region, close to the TARMsT2 attachment point, combined with our genetic and functional data, supports the hypothesis that this region could exhibit a valve-like activity. This activity is likely triggered by Dh31 or hCGRP following bacterial exposure. Notably, our previous research in adult *Drosophila* showed that Dh31/hCGRP secretion by EECs induces contractions of the visceral longitudinal muscle fibers, expelling pathogenic bacteria rapidly (*Benguettat et al., 2018*). However, in larvae, while Dh31/hCGRP likely induces muscle contractions, the ensuing action may manifest as the closure of the midgut junction, as evidenced by the observed retention of pathogenic bacteria. Importantly, in both larvae and adults, the same pathway and type of muscle fibers appear to be involved, as TARMsT2 are connected to longitudinal fibers (*Figure 6H'*; *LaJeunesse et al., 2010*). This finding is significant as it illustrates a conserved mechanism across developmental stages, albeit with different outcomes: expulsion of pathogens in adults and containment in larvae. In both cases, the overarching objective is the effective elimination of pathogens, demonstrating the versatility and adaptability of the *Drosophila* immune response.

Hence, our work shed lights on a yet to be anatomically characterized valve structure within the *Drosophila* larval gut, presenting a potential model for studying the functions and roles of mammalian pylori. Notably, CGRP (Calcitonin Gene-Related Peptide) secreting EECs have been identified in the mammalian pylorus, as highlighted in research by *Kasacka, 2009* and *Bulc et al., 2018*. Drawing parallels with mammalian stomachs, the pylorus is typically closed, opening only when the stomach becomes full. In exploring the functionality of the *Drosophila* valve we studied, we considered two hypotheses: one, where it operates similarly to its mammalian counterpart, closing by default and opening in response to a full stomach, and another, where it remains open by default and closes upon detection of infected food in the intestine. Our observations with the commensal bacteria *Lp* (*Video 4*) suggest an initial closure of the pylorus as food accumulates in the anterior part of the intestine, followed by its opening to allow the passage of non-pathogen-contaminated food. This indicates a dynamic and responsive mechanism in the *Drosophila* gut. In contrast, when *Drosophila* larvae encounter food contaminated with pathogenic bacteria, the valve seems to contract, effectively

blocking the passage of contaminated food into the anterior part of the intestine. This finding is significant as it not only reveals a unique physiological response in *Drosophila* larvae but also provides a basis for comparative studies with the mammalian gastrointestinal system, particularly in understanding the regulatory mechanisms governing pyloric function.

An intriguing question is why animals without a functional IMD pathway die from *Ecc* or *Bt* exposure while ROS production is still operating? Indeed, when the IMD pathway is disabled, bacteria are still confined in the anterior part of the gut, but are not effectively eliminated, resulting in larval death. This outcome strongly suggests that ROS alone are insufficient for bacterial eradication, even though they have been shown to damage and inhibit bacterial proliferation (*Benguettat et al., 2018*; *Ha et al., 2005*). However, the role of ROS in the immune response remains crucial since their timing of expression before antimicrobial peptides is likely a key factor for the efficiency of the immune response. Our findings, together with those from other studies, highlight the critical role of AMPs in fighting off virulent intestinal bacteria, particularly in scenarios where ROS activity is compromised or inadequate (*Ramond et al., 2021*; *Ryu et al., 2006*). Our findings emphasize the critical role of the ROS/TrpA1/Dh31 axis in effectively eradicating ingested pathogens. The rapid production of ROS following bacterial ingestion plays a pivotal role in closing the valve and retaining virulent bacteria in the anterior midgut. Notably, this process does not necessitate a transcriptional/translation response. In contrast, the production of antimicrobial peptides is a lengthier process, involving the transcriptional activation of *AMP* genes downstream of the IMD pathway, followed by their translation and secretion. The initial confinement of bacteria in the anterior midgut allows time for AMPs to be produced to eliminate the trapped bacteria, which is crucial for the organismal survival. This anterior bacterial blockage is important since when virulent bacteria reach the posterior midgut (in absence of ROS or in *TrpA1* or *Dh31* mutants) the larvae die despite a functional IMD pathway. A key question arises: Is the posterior midgut less equipped to combat bacteria? Interestingly, it has been previously shown in both *Drosophila* larvae and adults that while the posterior midgut can produce AMPs, it is predominantly dedicated to dampening the immune response, particularly through the production of amidases and the activity of the transcription factor Caudal, both that repressing AMP gene expression (*Bosco-Drayon et al., 2012*; *Hachfi et al., 2024*; *Ryu et al., 2008*). This permits an immune tolerance likely fostering the establishment of the commensal flora. Consistent with this, we observed that commensal bacteria like *Lp* transit from the anterior to the posterior midgut and persist there without compromising larval survival. Additionally, *Lp* inherent resistance to AMPs (*Arias-Rojas et al., 2023*) further underscores the idea that the anterior midgut serves as a checkpoint where certain bacteria are detained and eliminated, while others, like *Lp*, are permitted to pass through. When the compartmentalization mechanism is compromised, pathogens or pathobionts can spread into the posterior midgut, potentially eliciting an inadequate dampened immune response. Thus, the anterior midgut acts as a critical juncture in determining the fate of ingested bacteria, either leading to their elimination or allowing their passage to the posterior midgut, where a more tolerant immune environment prevails.

Understanding the practical application of this defense mechanism in the natural environment of *Drosophila melanogaster* larvae is crucial. In the wild, *Drosophila* adults are typically drawn to rotting fruits on which they lay eggs, exposing them and their progeny to a plethora of fungi and bacteria. Consequently, developing larvae feed and grow in these non-sterile conditions. In such environments, encountering pathogenic microbes is inevitable. Evasion or avoidance behavior has been documented as a potential strategy for dealing with pathogens (*Surendran et al., 2017*). This behavior might enable larvae to seek environments that will supposedly better sustain their survival. However, given the larvae constant consumption of their surrounding media in a race to reach pupation, ingestion of pathogen-contaminated food is a common risk. Under these circumstances, larvae have limited options prior to the activation of their innate immune response. Discriminating innocuous from potentially deleterious bacteria and then limiting the transit of the latter ones for subsequent elimination by AMPs, clearly benefits the host. Nonetheless, the effectiveness of this blockage strategy would be maximized if it were coordinated with evasion behaviors. Such coordination could prevent repeated engagement in this energy-intensive immune response, thus optimizing the larvae chances of reaching pupation successfully. In this context, it is interesting to note that the ROS/TrpA1/Dh31 axis is shared between larvae and adults to manage bacterial infection, but larvae block and kill the pathogens while adults eject them. If larvae eject their gut content, they may ingest it within

minutes while adults can fly to another fruit within seconds. This interplay between immune response and behavioral adaptation underlines the sophisticated strategies employed by *Drosophila* larvae to navigate their microbial-rich environment.

## Materials and methods

### Bacterial strain

We used the following strains: *Bacillus-thuringiensis*-GFP (*Bt*) (*Hachfi et al., 2024*) (the original strain, 4D22, is from the Bacillus Genetics Stock Center - https://www.bgsc.org/), *Erwinia carotovora* subsp. *carotovora*-GFP *15* (*Ecc*; *Basset et al., 2000*), *Escherichia coli* strain OP50-GFP (a slow growing and innocuous strain used to feed nematodes; *Eco*; gift from Jonathan Ewbank) and *Lactiplantibacillus plantarum*-GFP (*Lp*; gift from Renata Matos and François Leulier; *Storelli et al., 2018*). *Eco*, *Bt* and *Ecc* were grown on standard LB agar plates at 37 °C (*Eco*) or 30 °C (*Bt*, *Ecc*) and *Lp* was grown in MRS medium in anaerobic conditions at 37 °C for at least 18 hr. Importantly, we used vegetative cells of *Bt* without Cry toxins. Cry toxins are only produced during sporulation and are enclosed in a crystal within the spore. The *Bt* strain we used is 4D22 which have been deleted for the plasmids encoding for the Cry toxins. So, there is no Cry toxin in the *Bt*-GFP vegetative cells we used.

The solid medium used in our experiments was supplemented with antibiotics to ensure the selective growth of our bacterial strains as follows: *Ecc*: Spectinomycin at 100 µg/ml; *Bt*: Erythromycin at 10 µg/ml; *Lp*: Chloramphenicol at 10 µg/ml. Each bacterium was plated from glycerol stocks for each experiment. A single colony was used to prepare liquid cultures. Bacteria were inoculated in 500 ml of appropriate medium containing antibiotics as follows *Ecc*: Spectinomycin at 100 µg/ml; *Bt*: Erythromycin at 10 µg/ml; *Lp*: Chloramphenicol at 10 µg/ml. After overnight growth, the cultures were centrifuged for 15 min at 7500 rpm. Bacterial infectious doses were adjusted by measuring culture turbidity at an optical density of 600 nm. $OD_{600}$=100 for *Eco*, Lp and *Ecc* corresponds to $4.9.10^7$ CFU/µl. $OD_{600}$=100 for *Bt* corresponds to $1,5.10^7$ CFU/µl.

### FLY stocks

Flies were maintained at 25 °C on our standard fly medium (*Nawrot-Esposito et al., 2020*) with12:12 light/dark cycle. Fly stocks used in this study and their origins are as follows: Canton S (Bloomington #64349, the reference to compare with mutants and noted as control), $w^{1118}$ (Bloomington #5905), $w^1$ (Bloomington #145), *Oregon* (Bloomington #5), $TrpA1^1$ (Bloomington #26504), $Dh31^{KG09001}$ (Bloomington #16474), *Dh31-Gal4* (Bloomington #51988), $PGRP-LC^{\Delta E}$ (Bloomington #55713), $PGRP-LE^{112}$ (Bloomington #33055), $Dredd^{F64}$ (Gift from B. Charroux), $Relish^{E20}$ (Bloomington #55714), DJ752-Gal4 (Bloomington #8182), *ΔAMP14* (Gift from B. Lemaitre; *Carboni et al., 2022*), *yw;;Dpt-Cherry* (Gift from B. Charroux), *TrpA1*-Gal4/UAS-RFP (Gal4 is Bloomington #527593, UAS is Bloomington #27392), *Da*-Gal4 (Bloomington #55851), *Pros*-Gal4 (gift form B. Charroux), *Mex*-Gal4 (gift from B. Charroux), UAS-*Duox _IR* (Bloomington #38907), UAS-*Dh31_IR* (Bloomington#25925).

### Infection experiments

Oral infections were performed on mid-L3 larvae (3.5 days after egg laying). For each experiment, between 20 and 50 non-wandering L3 larvae raised at 25 °C were collected and washed in PBS (1 x). Bacterial pellets with starting OD 600 nm ranging from 300 to 400 were aligned with PBS1x at twice the desired final concentration. Then, they were mixed 1:1 with yeast 40% in PBS (1 x) and 500 µl of the infected food were added at the bottom of an empty plastic fly vial (VWR) before adding the larvae and sealing it with Parafilm. Then, the larvae were placed at 25 °C in the dark. After 60 min, the larvae were washed in PBS (1 x) and then counted for the presence of GFP-bacteria or for other analyses. For experiments involving fluorescent Dextran, 1.25 mg/mL of Dextran 4 kDa (Sigma ref 46944 for FITC-coupled and ref T1037 for TRITC-coupled) was added to the feeding medium. For experiments involving Uracil supplementation, Uracil (SIGMA U0750-5G) was added at 20 nM final to the feeding mixture containing *Lp*.

### Larvae dissection

After 60 min, the infected larvae were washed in PBS (1 x). Guts were dissected and fixed in formaldehyde 4% for 45 min, then washed twice in PBS (1 x) for 10 min. Guts were mounted between

poly-L-lysine (SIGMA P8920-100ML) coated slides and coverslips in Vectashield/DAPI (Vector Laboratories).

## Colony-forming unit (CFU) counting

Following the established protocol for infection experiments, animals were exposed to contaminated food for 1 hr. Subsequently, animals that had ingested bacteria were either immediately processed for the initial time-point analysis or transferred to a wet chamber for assessment at subsequent time-points. Importantly, since there was no further exposure to contaminated food beyond the 1 hr treatment period, the CFU assays we conducted measured the changes in the initial bacterial load within the larvae over time. Infected animals were washed in ethanol 70% for 30 s then rinsed in PBS (1 x). Guts were dissected in PBS (1 x) and homogenized with a micropestle in 200 µl of LB medium. Samples were serially diluted in LB medium and plated on LB agar plates overnight at 30 °C. The colonies forming unit (CFU) were counted the following day. CFU counting has been performed at 5 time points: 1 hr, 2 hr, 4 hr, 6 hr, and 8 hr after a 60-min intoxication (at least 20 larvae per point and 3 independent repeats).

## Mortality test

Oral infection of the larvae was performed as described above in *Infection experiments*. Larvae of the different genotypes fed 1 hr with *Bt*, *Lp,* or *Ecc* were quickly washed in 70% ethanol and then PBS (1 x). Only the larvae that have eaten (containing GFP bacteria in their intestine) were selected and put in a wet chamber for 18 hr at 25 °C. Mortality was evaluated at this time-point.

## DTT and CGRP feeding

Oral infection of the larvae was performed as described above in *Infection experiments*.

> DTT: nDTT was added to the food at a final concentration of 100 nM and larvae were fed during 60 min.
> hCGRP: hCGRP (Sigma #C0167) was resuspended in distilled water. Larvae were fed 1 hr as described above with a final hCGRP concentration of 400 µg/ml.

## Immunostaining

Dissected intestines were washed twice with PBS (1 x)–0.1% Triton X100 then incubated for 3 hr in the blocking solution (10% of fetal calf serum, 0.1% Triton X100, PBS 1 x). The blocking solution was removed and the primary antibodies added and incubated overnight à 4 °C in blocking solution. The following antibodies were used: mouse anti-Prospero (MR1A-c, Developmental Studies Hybridoma Bank DSHB) at 1:200 and rabbit anti-Dh31 (gift from Jan Veenstra and Michael Nitabach; *Kunst et al., 2014*; *Park et al., 2008*) at 1:500. Secondary antibodies used were anti-mouse Alexa647 (Invitrogen Cat# A-21235), anti-rabbit Alexa546 (Invitrogen Cat# A-11010). All secondary antibodies were used at 1:1000. Guts were mounted in Fluoroshield-DAPI mounting medium (Sigma F6057). Observations of GFP producing bacteria and of TRPA1 + cells were done using the native fluorescence without immunostaining. For microscopy involving actin staining, fluorescent phalloidin (Sigma P5282, 1/100) was added following the above protocol as if it was the primary antibody, but with a 10 min incubation time.

## RNAi experiments

All the tested animals were F1 obtained from a cross between parents possessing the Gal4 transgene and parents possessing the UAS-RNAi construction or from crosses between w- animals (genetic background of the Gal4 and UAS lines) and a transgenic line to serve as ctrl. The larvae were then fed with contaminated food as described above.

## RT-qPCR

Oral infection of the larvae was performed as described above in *Infection experiments*. Larvae of the different genotypes fed 1 hr with *Bt*, *Lp*, *Ecc* or yeast only were quickly washed in 70% ethanol and then PBS (1 x). Only the larvae that have eaten (containing GFP bacteria in their intestine, except for the yeast only condition) were selected and put in a wet chamber in the dark for 5 hr at 25 °C. RNA

from larval guts (n = 10 for each test and condition) was extracted with RNeasy Mini Kit (QIAGEN, cat. #74106). Quantitative real-time PCR, TaqMan, and SYBR Green analysis were performed as previously described in *Charroux et al., 2018*. The amount of mRNA detected was normalized to control rp49 mRNA values. Normalized data was used to quantify the relative levels of a given mRNA according to cycling threshold analysis (ΔCt). Control and experimental conditions were tested in the same 'run'. Each sample was normalized to its own rp49 control to take into account age-specific changes in gene expression. Results are presented as average and standard deviation of arbitrary CT of the tested gene from a minimum of three independent experiments. See the source data file for details. Primers used for RT-qPCR are:

> rp49: GACGCTTCAAGGGACAGTATCTG, AAACGCGGTTCTGCATGA
> Diptericin: GCTGCGCAATCGCTTCTACT, TGGTGGAGTGGGCTTCATG
> Attacin D: GTCACTAGGGTTCCTCAG, GCCGAAATCGGACTTG
> Drosomycin: CGTGAGAACCTTTTCCAATATGATG, TTCCACGACCACCAGCAT

## Images and movie acquisition

Images acquisition was performed at the microscopy platform of the Institut Sophia Agrobiotech (INRAE 1355-UCA-CNRS 7254-Sophia Antipolis) with the macroscope Zeiss AxioZoom V16 with an Apotome 2 or a Zeiss Axioplan Z1 with Apotome 2 microscope. Images were analyzed using ZEN and Photoshop softwares. Movie acquisitions were performed with the macroscope Zeiss AxioZoom V16 equipped with the Hamamatsu Flash 4LT Camera. Larvae were captured every 5 min. Dead larva images were acquired with a numeric Keyence VHX 2000 microscope.

## Data representation and statistical analyses

The Graphpad Prism 8 software was used for statistical analyses.

## CFU data analysis

the D'Agostino–Pearson test to assay whether the values are distributed normally was applied. As not all the data sets were considered normal, non-parametric statistical analysis such as non-parametric unpaired Mann–Whitney two-tailed tests was used for all the data presented.

## Blockage ratio and survival ratio datasets

as the values obtained from one larva are categorical data with a *Yes* or *No* value, we used the two-sided Fisher exact t-test and the 95% confidence interval to test the statistical significance of a possible difference between a test sample and the related control.

For all the quantitative assays, at least three independent experiments were performed and some were done in two different laboratories by more than one experimenter. The results from all the experiments were gathered and the total amount of larvae tested is indicated in the source data file. In addition, we do not show the average response from one experiment representative of the different biological replicates, but an average from all the data generated during the independent experiments in one graph.

## Acknowledgements

We are grateful to all members of the BES and DEB teams at the Institut Sophia Agrobiotech for fruitful discussions. We greatly thank Emilie Avazéri, Ambre Bigot, Elisa Di Lelio, Juliette Dubois, Marie-Paule Esposito and Gladys Gazelle for their technical support. We thank Bernard Charroux (at the IBDM Aix Marseille University) and Ambra MASUZZO (in the team of Richard Benton, Université de Lausanne) for pioneer observations and numerous discussions, François Leulier and Renata Matos for sharing bacterial and fly lines and Frank Schnorrer for discussions about alary muscles. FT was supported by the Lebanese Association for Scientific Research (LASER), the AJAJE association from Lebanon, and the Université Côte d'Azur (ATER). This work was supported by the French government through the UCAJEDI Investments in the Future project managed by the National Research Agency (ANR) with the reference number ANR-15-IDEX-01 and through the ANR-22-CE35-0006-01 (BaDAss) to AG. This work was supported by CNRS, ANR BACNEURODRO (ANR-17-CE16-0023-01),

Equipe Fondation pour la Recherche Médicale (EQU201603007783) and the ANR Pepneuron (ANR-21-CE16-0027) to JR and LK.

## Additional information

### Funding

| Funder | Grant reference number | Author |
|---|---|---|
| Agence Nationale de la Recherche | ANR-15-IDEX-01 | Fatima Tleiss<br>Olivier Pierre<br>Armel Gallet |
| Agence Nationale de la Recherche | ANR-22-CE35-0006-01 | Fatima Tleiss<br>Olivier Pierre<br>Armel Gallet |
| Agence Nationale de la Recherche | ANR-17-CE16-0023-01 | Martina Montanari<br>Romane Milleville<br>Julien Royet<br>C Leopold Kurz |
| Fondation pour la Recherche Médicale | EQU201603007783 | Martina Montanari<br>Romane Milleville<br>Julien Royet<br>C Leopold Kurz |

The funders had no role in study design, data collection and interpretation, or the decision to submit the work for publication.

### Author contributions

Fatima Tleiss, Martina Montanari, Olivier Pierre, Conceptualization, Data curation, Formal analysis, Investigation, Methodology, Writing – original draft; Romane Milleville, Conceptualization, Data curation, Formal analysis, Investigation, Methodology, Writing – review and editing; Julien Royet, Dani Osman, Armel Gallet, Conceptualization, Resources, Data curation, Formal analysis, Supervision, Funding acquisition, Validation, Investigation, Methodology, Writing – original draft, Project administration, Writing – review and editing; C Leopold Kurz, Conceptualization, Resources, Data curation, Formal analysis, Supervision, Validation, Investigation, Methodology, Writing – original draft, Project administration, Writing – review and editing

### Author ORCIDs

Julien Royet ⓘ https://orcid.org/0000-0002-5671-4833
Dani Osman ⓘ https://orcid.org/0000-0003-3880-3098
Armel Gallet ⓘ http://orcid.org/0000-0002-2054-4780
C Leopold Kurz ⓘ https://orcid.org/0000-0001-7081-3208

Reviewer #1 (Public review): https://doi.org/10.7554/eLife.98716.3.sa1
Reviewer #2 (Public review): https://doi.org/10.7554/eLife.98716.3.sa2
Author response https://doi.org/10.7554/eLife.98716.3.sa3

## Additional files

### Supplementary files
• MDAR checklist

• Source data 1. The source data file contains the genotypes, raw data and statistical analyses for all the data presented in the figures and figure supplements.

### Data availability
Genotypes, raw data and statistics are available at: https://doi.org/10.6084/m9.figshare.25018352.

The following dataset was generated:

| Author(s) | Year | Dataset title | Dataset URL | Database and Identifier |
|---|---|---|---|---|
| Tleiss F, Montanari M, Pierre O, Royet J, Osman D, Gallet A, Kurz LC | 2024 | Tleiss et al_eLife 2024_ Source Data File | https://doi.org/10.6084/m9.figshare.25018352 | figshare, 10.6084/m9.figshare.25018352 |

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
