## [Editor Report · eLife Assessment]

This article describes a novel mechanism allows *Drosophila* to combat enteric pathogens while also preserving the beneficial indigenous microbiota. The authors provide **compelling** evidence that oral infection of *Drosophila* larvae by pathogenic bacteria activate a valve that traps the intruders in the anterior midgut, allowing them to be killed by antimicrobial peptides. This is an **important** finding revealing a new mechanism of host defense in the gut of insects.

---

## [Referee Report · Reviewer #1 (Public review)]

Tleiss et al. demonstrate that while commensal Lactiplantibacillus plantarum freely circulate within the intestinal lumen, pathogenic strains such as *Erwinia carotovora* or Bacillus thuringiensis are blocked in the anterior midgut where they are rapidly eliminated by antimicrobial peptides. This sequestration of pathogenic bacteria in the anterior midgut requires the Duox enzyme in enterocytes, and both TrpA1 and Dh31 in enteroendocrine cells. This effect induces muscular muscle contraction, which is marked by the formation of TARM structures (thoracic ary-related muscles). This muscle contraction-related blocking happens early after infection (15mins). On the other side, the clearance of bacteria is done by the IMD pathway possibly through antimicrobial peptide production while it is dispensable for the blockage. Genetic manipulations impairing bacterial compartmentalization result in abnormal colonization of posterior midgut regions by pathogenic bacteria. Despite a functional IMD pathway, this ectopic colonization leads to bacterial proliferation and larval death, demonstrating the critical role of bacteria anterior sequestration in larval defense.

In general, this fundamentally important study reveals unique mechanisms in the gut immunity of *Drosophila* larvae. It also describes a previously understudied structure, TARM, which may play a crucial role in this process. This significant work substantially advances our understanding of pathogen clearance by identifying a new mode of pathogen eradication from the insect gut. The evidence supporting the authors' claims is compelling, and the study opens new avenues for future research in gut immunity.

---

## [Referee Report · Reviewer #2 (Public review)]

Summary:

This article describes a novel mechanism of host defense in the gut of *Drosophila* larvae. Pathogenic bacteria trigger the activation of a valve that blocks them in the anterior midgut where they are subjected to the action of antimicrobial peptides. In contrast, beneficial symbiotic bacteria do not activate the contraction of this sphincter and can access the posterior midgut, a compartment more favorable to bacterial growth.

Strengths:

The authors decipher the underlying mechanism of sphincter contraction, revealing that ROS production by Duox activates the release of DH31 by enteroendocrine cells that stimulate visceral muscle contractions. Use of mutations affecting the Imd pathway or lacking antimicrobial peptides reveals their contribution to pathogen elimination in the anterior midgut.

Weaknesses:

The mechanism allowing the discrimination between commensal and pathogenic bacteria remains unclear.

---

## [Author Response]

The following is the authors’ response to the original reviews.

**Reviewer #1 (Public Review):**
Tleiss et al. demonstrate that while commensal Lactiplantibacillus plantarum freely circulate within the intestinal lumen, pathogenic strains such as *Erwinia carotovora* or Bacillus thuringiensis are blocked in the anterior midgut where they are rapidly eliminated by antimicrobial peptides. This sequestration of pathogenic bacteria in the anterior midgut requires the Duox enzyme in enterocytes, and both TrpA1 and Dh31 in enteroendocrine cells. This effect induces muscular muscle contraction, which is marked by the formation of TARM structures (thoracic ary-related muscles). This muscle contraction-related blocking happens early after infection (15mins). On the other side, the clearance of bacteria is done by the IMD pathway possibly through antimicrobial peptide production while it is dispensable for the blockage. Genetic manipulations impairing bacterial compartmentalization result in abnormal colonization of posterior midgut regions by pathogenic bacteria. Despite a functional IMD pathway, this ectopic colonization leads to bacterial proliferation and larval death, demonstrating the critical role of bacteria anterior sequestration in larval defense.This important work substantially advances our understanding of the process of pathogen clearance by identifying a new mode of pathogen eradication from the insect gut. The evidence supporting the authors' claims is solid and would benefit from more rigorous experiments.(1) The authors performed the experiments on *Drosophila* larvae. I wonder whether this model could extend to adult flies since they have shown that the ROS/TRPA1/Dh31 axis is important for gut muscle contraction in adult flies. If not, how would the authors explain the discrepancy between larvae and adults?

We have linked the adult phenotype to the larval model to explore the ROS/TrpA1/Dh31 axis in both contexts. As highlighted in the discussion, however, there are key behavioral differences between larvae and adult flies. Unlike larvae, which remain in the food environment, adult flies have the ability to move away. This difference could impact the relevance of gut muscle contraction and bacterial clearance mechanisms between the two stages. Specifically, in larvae, the rapid ejection of gut contents due to muscle contraction poses a unique risk: larvae may inadvertently re-ingest the expelled material within minutes, which could influence their immune defenses. We have clarified this distinction and our hypothesis in the final section of the discussion, as it emphasizes the adaptive nature of this mechanism in larvae.

(2) The authors performed their experiments and proposed the models based on two pathogenic bacteria and one commensal bacterial at a relatively high bacterial dose. They showed that feeding Bt at 2X1010 or Ecc15 at 4X108 did not induce a blockage phenotype.I wonder whether larvae die under conditions of enteric infection with low concentrations of pathogenic bacteria.

To address this, we have provided new data (Movie 5), in which larvae were fed a lower dose of Bt-GFP at 1.3 × 10^10 CFU/mL. In this video, we observe that when larvae ingest fewer bacteria, no blockage occurs, and the bacteria are able to reach the posterior midgut. As the bacterial load is lower, the fluorescence signal is weaker, but the movie clearly shows the excretion of bacteria. Importantly, under these conditions, no larval death was observed. These findings suggest that below a certain bacterial threshold, the pathogenicity is insufficient to: (1) trigger the blockage response, and (2) kill the larvae. In such cases, bacteria are likely eliminated through normal peristaltic movements rather than through the blockage mechanism described in our study.

If larvae do not show mortality, what is the mechanism for resisting low concentrations of pathogenic bacteria?

As mentioned in our previous response, we hypothesize that the larvae’s ability to resist low concentrations of pathogenic bacteria is likely due to being below the threshold of virulence. At lower bacterial doses, the pathogenic load is insufficient to trigger the blockage mechanism or cause larval death. In these cases, it is probable that classical peristaltic movements of the gut efficiently eliminate the bacteria, preventing them from colonizing the posterior midgut or causing significant harm. Thus, the larvae rely on standard gut motility and immune mechanisms, rather than the blockage response, to clear lower doses of bacteria.

Why is this model only applied to high-dose infections?

The reason this model primarily applies to high-dose infections is that lower concentrations of pathogenic bacteria do not trigger the blockage mechanism. As we mentioned in the manuscript, for low bacterial concentrations, where the GFP signal remains detectable, wild-type larvae are still able to resist live bacteria in the posterior part of the intestine.

Regarding the bacterial doses used in our experiments, it's important to clarify that we calculate the bacterial load based on colony-forming units (CFU). In our setup, there are approximately 5 × 10^4 CFU per midgut. For each experiment, we prepare 500 µl of contaminated medium containing 4 × 10^10 CFU. Fifty larvae are placed into this 500 µl of medium, meaning each larva ingests around 5 × 10^4 CFU within one hour of feeding.

This leads us to two key points:

(1) Continuous feeding might trigger the blockage response even at lower doses, as extended exposure to bacteria could lead to higher accumulation within the gut.

(2) Other defense mechanisms, such as the production of reactive oxygen species (ROS) or classical peristaltic movements, could be sufficient to eliminate lower bacterial doses (around 10^3 CFU or below).

We also refer to the newly provided Movie 5, where larvae fed with Bt-GFP at 1.3 × 10^10 CFU/mL show no blockage at low ingestion levels and successfully eliminate the bacteria.

(3) The authors claim that the lock of bacteria happens at 15 minutes while killing by AMPs happens 6-8 hours later.

Our CFU data indicate that it’s after 4 to 6 hours that the quantity of bacteria decreases. We fixed this in the text.

What happened during this period?

During the 4 to 6-hour period, several defense mechanisms are activated. ROS play a bacteriostatic and bacteriolytic role, helping to control bacterial growth. Concurrently, the IMD pathway is activated, leading to the transcription, translation, and secretion of antimicrobial peptides. These AMPs exert both bacteriostatic and bacteriolytic effects, contributing to the eventual clearance of the pathogenic bacteria.

More importantly, is IMD activity induced in the anterior region of the larval gut in both Ecc15 and Bt infection at 6 hours after infection?

We have provided new data (Supplementary Figure 6) that includes RT-qPCR analysis of the whole larval gut in wt, TrpA1- and Dh31- genetic background after feeding with Lp, Ecc15, Bt, or yeast only. We monitored the expression of three different AMP-encoding genes and found that while AMP expression varied depending on the food content, there were no significant differences between the genotypes tested.

Additionally, we included new imaging data (Supplementary Figure 11) from AMP reporter larvae (*Dpt*-Cherry) fed with fluorescent Lp or Bt. In larvae infected with Bt, which is blocked in the anterior part of the gut, the *dpt* gene is predominantly induced in this region, indicating strong IMD pathway activity in response to Bt infection. Conversely, in larvae fed with Lp-GFP, the *Dpt*-Cherry reporter shows weak expression in the anterior midgut, and is barely detectable in the posterior midgut where Lp-GFP establishes itself. This aligns with previous findings by Bosco-Drayon et al. (2012), which demonstrated low AMP expression in the posterior midgut due to the presence of negative regulators of the IMD pathway, such as amidases and Pirk.

Are they mostly expressed in the anterior midgut in both bacterial infections? Several papers have shown quite different IMD activity patterns in the *Drosophila* gut. Zhai et al. have shown that in adult *Drosophila*, IMD activity was mostly absent in the R2 region as indicated by dpt-lacZ. Vodovar et al. have shown that the expression of dpt-lacZ is observable in proventriculus while Pe is not in the same region. Tzou et al. showed that Ecc15 infection induced IMD activity in the anterior midgut 24 hours after infection.

Based on our new data (Supplementary Figure 11), we observe that *Dpt*-RFP expression is primarily localized in the anterior midgut and likely in the beginning of acidic region in larvae infected with Bt, Ecc and Lp.

Using TrpA1 and Dh31 mutants, the authors found both Ecc15 and Bt in the posterior midgut. Why are they not evenly distributed along the gut?

We observe that bacteria are not evenly distributed along the gut in wild-type larvae as well, with LP. This suggests that the transit time in the anterior part of the gut may be relatively short due to active peristaltism, which would make this region function as a "checkpoint" for bacteria that are not supposed to be blocked. Indeed, we confirmed that peristaltism is active during our intoxication experiments, which could explain the rapid movement of bacteria through the anterior midgut.

In contrast, bacteria tend to remain longer in the posterior midgut, which corresponds to the absorptive functions of intestinal cells in this region. This would explain why we observe more bacteria in the posterior midgut for *Lp* in control larvae and for *Ecc15* and *Bt* in the TrpA1- and Dh31- mutants. Although a few bacteria are still found in the anterior midgut, they are consistently in much lower numbers compared to the posterior, as shown in Figures 1A and 3A of our manuscript.

Last but not least, does the ROS/TrpA1/Dh31 axis affect AMP expression?

We investigated whether the ROS/TrpA1/Dh31 axis influences AMP expression by performing RT-qPCR on the whole gut of larvae in wild-type, TrpA1-, and Dh31- genetic backgrounds. Larvae were fed with Lp, Ecc, Bt, or yeast (new data: Supplementary Figure 6). We monitored the expression of three different AMP-encoding genes and found that while AMP expression varied depending on the food content, there were no significant differences in AMP expression between the different genotypes.

Additionally, we provide imaging data from AMP reporter larvae (*pDpt*-Cherry) fed with fluorescent Lp or Bt (new data: Supplementary Figure 11). These results further confirm that the ROS/TrpA1/Dh31 axis does not significantly affect AMP expression in our experimental conditions.

(4) The TARM structure part is quite interesting. However, the authors did not show its relevance in their model. Is this structure the key-driven force for the blocking phenotype and killing phenotype?

We agree that the TARM structures are a fascinating aspect of this study and acknowledge the interest in their potential role in the blocking and killing phenotypes. While we are keen to explore the specific contributions of these structures during bacterial intoxication, the current genetic tools available for manipulating TARMs target both TARM T1 and T2 simultaneously, as demonstrated by Bataillé et al., 2020 (Fig. 2). Of note, these muscles are essential for proper gut positioning in larvae, and their absence leads to significant defects in food intake and transit, which would confound the results of our intoxication experiments (see Fig. 6 from Bataillé et al., 2020).

Therefore, while TARMs are likely involved in these processes, the current limitations in selectively targeting them prevent us from definitively testing their role in bacterial blocking and killing at this stage. We hope to address this in future studies as more refined genetic tools become available.

Is the ROS/TrpA1/Dh31 axis required to form this structure?

To determine whether the ROS/TrpA1/Dh31 axis is required for the formation of TARM structures, we examined larval guts from control, TrpA1-, and Dh31- mutant backgrounds. Our new data (Supplementary Figure 8) show that the TARM T2 structures are still present in the mutants, indicating that the formation of these structures does not depend on the ROS/TrpA1/Dh31 axis.

**Reviewer #2 (Public Review):**
This article describes a novel mechanism of host defense in the gut of *Drosophila* larvae. Pathogenic bacteria trigger the activation of a valve that blocks them in the anterior midgut where they are subjected to the action of antimicrobial peptides. In contrast, beneficial symbiotic bacteria do not activate the contraction of this sphincter, and can access the posterior midgut, a compartment more favorable to bacterial growth.Strengths:The authors decipher the underlying mechanism of sphincter contraction, revealing that ROS production by Duox activates the release of DH31 by enteroendocrine cells that stimulate visceral muscle contractions. The use of mutations affecting the Imd pathway or lacking antimicrobial peptides reveals their contribution to pathogen elimination in the anterior midgut.Weaknesses:The mechanism allowing the discrimination between commensal and pathogenic bacteria remains unclear.

Based on our findings, we hypothesize that ROS play a crucial role in this discrimination process, with uracil release by pathogenic or opportunistic bacteria potentially serving as a key signal.

To test whether uracil could trigger this discrimination, we conducted experiments where Lp was supplemented with uracil. However, our results show that uracil supplementation alone was not sufficient to induce the blockage response (new data: Supplementary Figure 5). This suggests that while uracil may be a factor in bacterial discrimination, it is likely not the sole trigger, and additional bacterial factors or signals may be required to activate the blockage mechanism.

The use of only two pathogens and one symbiotic species may not be sufficient to draw a conclusion on the difference in treatment between pathogenic and symbiotic species.

To address this concern, we performed additional intoxication experiments using *Escherichia coli* OP50, a bacterium considered innocuous and commonly used as a standard food source for *C. elegans* in laboratory settings. The results, presented in our updated data (new data: Fig 1B), show that *E. coli* OP50, despite being from the same genus as Ecc, does not trigger the blockage response. This further supports our conclusion that the gut’s discriminatory mechanism is specific to pathogenic bacteria, and not merely based on bacterial genus.

We can also wonder how the process of sphincter contraction is affected by the procedure used in this study, where larvae are starved. Does the sphincter contraction occur in continuous feeding conditions? Since larvae are continuously feeding, is this process physiologically relevant?

In our intoxication protocol, the larvae are exposed to contaminated food for 1 hour, during which the blockage ratio is quantified. Since this period involves continuous feeding with the contaminated food, we do not consider the larvae starved during the quantification process. Our observations show differences in the blockage response depending on the bacterial contaminant and the genetic background of the host. Additionally, we were able to trigger the blocking phenomenon using exogenous hCGRP.

Regarding the experimental setup for movie observations, it is true that larvae are immobilized on tape in a humid chamber, which is not a fully physiological context. However, in the new movie we provide (Movie 3), co-treatment with fluorescent Dextran (Red) and fluorescent Bt (Green) shows that both are initially blocked, followed by the posterior release of Dextran once the bacterial clearance begins.

Furthermore, to address the question of continuous exposure, we extended the exposure period to 20 hours instead of 1 hour. Even after prolonged exposure, we observed that pathogens are still blocked in the anterior part of the gut (new data: Supplementary Figure 2B). This supports the physiological relevance of the sphincter contraction and its ability to function under continuous feeding conditions.

**Reviewer #1 (Recommendations For The Authors):**
(1) The authors performed the experiments on *Drosophila* larvae. I wonder whether this model could extend to adult flies since they have shown that the ROS/TRPA1/Dh31 axis is important for gut muscle contraction in adult flies. If not, how would the authors explain the discrepancy between larvae and adults?

We link the adult phenotype to the one we describe in larvae in order to have the candidate approach toward the ROS/TrpA1/Dh31 axis. As we already mention in the discussion, while larvae stay in the food, adult flies can go away. If larvae eject their gut content, they may ingest it within minutes. We clarify our idea in the last part of the discussion.

(2) The authors performed their experiments and proposed the models based on two pathogenic bacteria and one commensal bacterial at a relatively high bacterial dose. They showed that feeding Bt at 2X1010 or Ecc15 at 4X108 did not induce a blockage phenotype.I wonder whether larvae die under conditions of enteric infection with low concentrations of pathogenic bacteria.

Video provided with Bt-GFP 1.3 10^10 CFU/mL (new data: Movie 5). When larvae eat less, there is no blockage and bacteria can reach the posterior midgut. Note that the fluorescence is weak due to the low amount of bacteria ingested. The movie shows an excretion of the bacteria. There is also no death of the larvae. Together these results suggest that below a given threshold, the virulence of the bacteria is too weak to (i) trigger a blockage and 2/ kill the larva. The bacteria are likely eliminated through classical peristaltism.

If larvae do not show mortality, what is the mechanism for resisting low concentrations of pathogenic bacteria?

Maybe we are below the threshold of virulence. See our response just above.

Why is this model only applied to high-dose infections?

As mentioned in the manuscript, lower concentrations do not trigger the blockage and for lower concentrations with a GFP signal still detectable, wild-type animals resist the presence of live-bacteria within the posterior part of the intestine.

About the doses, the CFU should be considered. Indeed, there are around 5.10^4 CFU per midgut. In our experimental procedure we calculate the amount of bacteria for 500 µl of contaminated medium (i.e. 4.10^10 CFU/500µl of medium). Then around 50 larvae were deposited in the 500µl of contaminated media. In this condition, one larva ingests 5.10^4 CFU. Moreover, larvae are only fed for 1h.

So 1/ continuous feeding may also trigger locking even at lower doses and 2/ the other mechanisms of defenses (such as ROS) or peristalsis may be sufficient to eliminate lower doses (i.e. 10^3 CFU or below). See the new movie 5 we provide with *Bt-GFP 1.3 10^10 CFU/mL*

(3) The authors claim that the lock of bacteria happens at 15 minutes while killing by AMPs happens 6-8 hours later.

Our CFU data indicate that it’s after 4 to 6 hours that the quantity of bacteria decreases. We fixed this in the text.

What happened during this period?

ROS activity (bacteriostatic and bacteriolytic), IMD activation, AMP transcription, translation, secretion and bacteriostatic as well as bacteriolytic activity.

More importantly, is IMD activity induced in the anterior region of the larval gut in both Ecc15 and Bt infection at 6 hours after infection?

We provide new data for larval whole gut RT-qPCR data in wt, TrpA1- and Dh31- genetic background fed with Lp or Ecc or Bt or yeast only (new data: SUPP6). We monitored 3 different AMP-encoding genes and found differences related to the food content, but no differences between genotypes. In addition, we provide images from AMP reporter animals (Dpt-Cherry) fed with fluorescent Lp or Bt (new data: SUPP11) showing that with Bt blocked in the anterior part of the intestine, the *dpt* gene is mainly induced in this area. Note that in the larva infected with Lp-GFP, the Dpt-Cherry reporter is weakly expressed in the anterior midgut. In the posterior midgut, the place where Lp-GFP is established, Dpt-Cherry is barely detectable. This observation is in line with the previous observation made by Bosco-Drayon et al., (2012) demonstrating the low level of AMP expression in the posterior midgut due to the expression of the IMD negative regulators such as *amidases* and *pirk*. In the larva infected with Bt-GFP, note the obvious expression of DptCherry in the anterior midgut colocalizing with the bacteria (new data: SUPP11).

Are they mostly expressed in the anterior midgut in both bacterial infections? Several papers have shown quite different IMD activity patterns in the *Drosophila* gut. Zhai et al. have shown that in adult *Drosophila*, IMD activity was mostly absent in the R2 region as indicated by dpt-lacZ. Vodovar et al. have shown that the expression of dpt-lacZ is observable in proventriculus while Pe is not in the same region. Tzou et al. showed that Ecc15 infection induced IMD activity in the anterior midgut 24 hours after infection.

In ctrl animals fed Bt, Ecc and Lp we see Dpt-RFP in anterior midgut and likely in the beginning of acidic region. See the new data: SUPP11 images provided for the previous remark.

Using TrpA1 and Dh31 mutants, the authors found both Ecc15 and Bt in the posterior midgut. Why are they not evenly distributed along the gut?

Same is true with Lp in wt; not evenly distributed. As if the transit time in the anterior part is very short due to peristaltism which would fit for a check point area if you’re not supposed to be blocked. Indeed, peristaltism is active during our intoxications. Then, it stays longer in the posterior part, fitting with the absorptive skills of the intestinal cells in this area. With Lp in ctrl or Ecc and Bt in TrpA1- and Dh31- mutants, there are always a few in the anterior midgut but always much less compared to the posterior. See our figure 1A and 3A.

Last but not least, does the ROS/TrpA1/Dh31 axis affect AMP expression?

We provide larval whole gut RT-qPCR data in wt, TrpA1- and Dh31- genetic background fed with Lp or Ecc or Bt or yeast only (new data: SUPP6). We monitored 3 different AMPencoding genes and found differences related to the food content, but no differences between genotypes. In addition, we provide images from AMP reporter animals (pDptCherry) fed with fluorescent Lp or Bt, (new data: SUPP11).

(4) The TARM structure part is quite interesting. However, the authors did not show its relevance in their model. Is this structure the key-driven force for the blocking phenotype and killing phenotype?

Indeed, we would like to explore the roles of these structures and the putative requirement upon bacterial intoxication using some driver lines developed by the team that studied these muscles in vivo. However, the genetic tools currently available will target TARMsT1 and T2 at the same time. See Fig 2 form Bataillé et al, . 2020. Moreover, these TARMs are, at first, crucial for the correct positioning of the gut within the larvae and their absence lead to a global food intake and transit defect that will bias the outcomes of our intoxication protocol (see fig 6 from Bataillé et al,. 2020).

Is the ROS/TrpA1/Dh31 axis required to form this structure?

We provide images of larval guts from ctrl, TrpA1 and Dh31 mutants demonstrating the presence of the TARMs T2 structures despite the mutations (new data: SUPP8). In addition, we provide representative movies of peristalsis in intestines of Dh31 mutants fed or not with Ecc to illustrate that muscular activity is not abolished (new data: Movie 9 and Movie 10).

Minor points:(1) Why not use the Pros-Gal4/UAS-Dh31 strain in Figure 3B in addition to hCGRP?

We opted for exogenous hCGRP addition because it allowed us precise timing control over Dh31 activation. Overexpression of Dh31 from embryogenesis or early larval stages could have significant and unintended effects on intestinal physiology, potentially confounding the results. While temporal control using TubG80ts could be an alternative, our focus was on identifying the specific cells responsible for the phenomenon.

To achieve this, we perturbed Dh31 production via RNAi, specifically targeting a limited number of enteroendocrine cells (EECs) using the DJ752-Gal4 driver, as described by Lajeunesse et al., 2010. Our new data (Supplementary Figure 4) demonstrate that Dh31 expression in this subset of cells is indeed necessary for the blockage phenomenon.

(2) Section title (line 287) refers to mortality, but no mortality data is in the figure.

We agree that the title referenced mortality, whereas no mortality data was presented in this section. We have updated the title to better reflect the data discussed in this part of the manuscript.

(3) It may be better to combine ROS-related contents in the same figure.

While it is technically feasible to consolidate the ROS-related content into one figure, doing so would require splitting essential data, such as the *Gal4* controls for the RNAi assays and parts of the survival phenotype data. We believe that the current structure of the study, which first explores the molecular aspects of the phenomenon and then demonstrates its relevance to the animal’s survival, provides a clearer and more logical flow. For these reasons, we prefer to maintain the current figure layout.

**Reviewer #2 (Recommendations For The Authors):**
Major recommendation(1) Other wild-type backgrounds should be added (including the w Drosdel background of the AMP14 deficient flies) to check the robustness of the phenotype.

To address the concern regarding the robustness of the phenotype across different wildtype backgrounds, we have tested additional genetic backgrounds, including w1, the isogenized w1118 and Oregon animals.

The results (new data: Figure 1C) demonstrate that Lp is able to transit freely to the posterior part of the intestine in all backgrounds, while Ecc and Bt are blocked in the anterior part. These findings confirm the robustness of the phenotype across different wildtype strains.

(2) Although we recognize that this may be limited by the number of GFP-expressing species, other commensal and pathogenic bacteria should be tested in this assay (e.g. *E. faecalis* and Acetobacter).

We performed new intoxication experiments using *Escherichia coli* OP50, a wellestablished innocuous bacterial strain. The data, presented in Figure 1B (new data), show that *E. coli* OP50, despite being from the same genus as Ecc, does not trigger the blockage response. This further supports our hypothesis that the blockage phenomenon is specific to pathogenic bacteria and not simply related to the bacterial genus.

(3) It is important to test whether sphincter closure also occurs in continuous feeding conditions. This does not mean repeating all the experiments but just shows that this mechanism can take place in conditions where larvae are kept in a vial with food.

While the movies we provide involve larvae immobilized on tape in a humid chamber, which is not a fully physiological context, we now provide new data (Movie 3) showing that, after co-treatment with fluorescent Dextran (Red) and fluorescent Bt (Green), both substances are initially blocked in the anterior midgut. Later, the dextran is released posteriorly once bacterial clearance has begun.

Additionally, we extended the feeding period in our experiments from 1 hour to 20 hours to simulate more continuous exposure to contaminated food. Even under these prolonged conditions, we observed that pathogens are still blocked in the anterior part of the gut (new data: Supplementary Figure 2B). This confirms that the sphincter mechanism can function in continuous feeding conditions as well.

(4) What are the molecular determinants discriminating innocuous from pathogenic bacteria? Addressing this point will increase the impact of the article. The fact that Relish mutants have normal valve constriction suggests that peptidoglycan recognition is not involved. Is there a sensing of pathogen virulence factors?

Our data suggest that uracil could be a key molecular determinant in discriminating between innocuous and pathogenic bacteria, as previously described by the W-J Lee team in several studies on adult *Drosophila*. However, in our experiments, exogenous uracil addition using the blue dye protocol (Keita et al., 2017) did not induce any significant changes in the larvae. Similarly, uracil supplementation in adult flies failed to trigger the *Ecc* expulsion and gut contraction phenotype, as reported by Benguettat et al., 2018.

To further investigate this, we tested the addition of uracil during Lp-GFP intoxication. In these experiments, we did not observe any blockage of *Lp* (new data: Supplementary Figure 5). These results suggest that uracil might not be the sole trigger for the blockage response, or we may not be providing uracil exogenously in the most effective way. Alternatively, there could be other pathogen-specific virulence factors that contribute to this discrimination mechanism.

To address this question, the authors should infect larvae with Ecc15 evf- mutants or Ecc15 lacking uracil production.

Thank you for your suggestion to use *Ecc15* evf- mutants or *Ecc15* lacking uracil production to explore the role of uracil in bacterial discrimination. While we have provided some data using uracil supplementation (new data: Supplementary Figure 5), we agree that testing mutants like *PyrE* would be an important next step. Unfortunately, we currently lack access to fluorescent *PyrE* or *Ecc15* evf- mutants.

We are planning to address this by developing a new protocol involving fluorescent beads alongside bacteria. This approach will allow us to test several bacterial strains in parallel and better define the size threshold of the valve. However, we do not have the relevant data yet, but this will be a key focus of our future work.

Similarly, does feeding heat-killed Ecc15 or Bt induce sequestration in the anterior midgut (larvae may be fed dextran-FITC at the same time to track bacteria)?

Unfortunately, in our attempts to test heat-killed or ethanol-killed fluorescent *Ecc15* for these experiments, we encountered an issue: while we were able to efficiently kill the bacteria, we lost the GFP signal required to track their position in the gut. This made it challenging to assess whether sequestration in the anterior midgut occurs with non-viable bacteria.

Is uracil or Bt toxin feeding sufficient to induce valve closure?

As previously mentioned, uracil is a strong candidate for bacterial discrimination, and we have tested its role by adding exogenous uracil during Lp-GFP intoxication. However, in these experiments, *Lp* was not blocked (new data: Supplementary Figure 5). This suggests that uracil alone may not be sufficient to induce valve closure, or it may not be the only factor involved. It is also possible that our method of exogenous uracil supplementation may not be effectively mimicking the endogenous conditions.

Regarding Bt, we used vegetative cells without Cry toxins in our experiments. Cry toxins are only produced during sporulation and are enclosed in crystals within the spore. The Bt strain we used, 4D22, has been deleted for the plasmids encoding Cry toxins. As a result, there were no Cry toxins present in the Bt-GFP vegetative cells used in our assays. This has been clarified in the Materials and Methods section of the manuscript.

Would Bleomycin induce the same phenotype?

Indeed, Bleomycin, as well as paraquat, has been shown to damage the gut and trigger intestinal cell proliferation in adult *Drosophila* through mechanisms involving TrpA1. Testing whether Bleomycin induces a similar phenotype in larvae would indeed be interesting.

However, one challenge we face in our intoxication protocol is that larvae tend to stop feeding when chemicals are added to their food mixture. We encountered similar difficulties in our DTT experiments, which were challenging to set up for this reason. Consequently, we aim to avoid approaches that might impair the general feeding activity of the larvae, as it can significantly affect the outcomes of our experiments.

Could this process of sphincter closure be more related to food poisoning?

If gut damage were the primary trigger for sphincter closure, we would indeed expect the blockage phenomenon to occur later following bacterial exposure. However, in our experiments, we observe the blockage occurring early after bacterial contact, suggesting that damage may not be the main trigger for this response.

That said, we have not yet tested bacterial mutants lacking toxins, nor have we tested a direct damaging agent such as Bleomycin, as proposed. These would be valuable future experiments to explore the potential role of gut damage more thoroughly in this process.

(5) Is Imd activation normal in trpA1 and DH31 mutants? The authors could use a diptericin reporter gene to check if Diptericin is affected by a lack of valve closure in trpA1.

To address this, we performed RT-qPCR on whole larval guts from wt, TrpA11 and Dh31KG09001 genetic background. Larvae were fed with Lp, Ecc, Bt or yeast only (new data: SUPP6). We monitored the expression of three different AMP-encoding genes and found that while AMP expression varied depending on the food content, there were no significant differences in AMP expression between the genotypes.

Additionally, we provide imaging data from AMP reporter animals (*pDpt*-Cherry) in a wildtype background, fed with fluorescent Lp or Bt (new data: Supplementary Figure 11). These images also support the conclusion that *Diptericin* expression is not significantly affected by a lack of valve closure in *trpA1* and *Dh31* mutants.

(6) Are the 2-6 DH31 positive cells the same cells described by Zaidman et al., Developmental and Comparative Immunology 36 (2012) 638-647.

The cells identified as hemocytes in the midgut junctions by Zaidman et al. are likely the same cells we describe in our study, as they are located in the same region and are *Dh31* positive. We have added a reference to this paper and included lines in the manuscript acknowledging this connection.

Although confirming whether these cells are *Hml+*, *Dh31+*, and *TrpA1+* would clarify their exact identity, this falls outside the scope of our current study. However, the possibility that these cells play a role in physical barrier immunity and also possess a hemocyte identity is indeed intriguing, and we hope future research will explore this further.

Minor points(1) The mutations should be appropriately labelled with the allele name.

This has been fixed in the main text, in Fig Legends, and in figures.

(2) Line 230-231: the sentence is unclear to me.

We simplified the sentence and do not refer to the expulsion in larvae.

(3) Discussion: although the discussion is already a bit long, it would be interesting to see if this process is likely to happen/has been described in other insects (mosquito, Bactrocera, ...).

We reviewed the available literature but were unable to find specific examples describing the blockage phenomenon in other insects. Most studies we found focused on symbiotic bacteria rather than pathogenic or opportunistic bacteria. However, as mentioned in our manuscript, the anterior localization of opportunistic or pathogenic bacteria has been observed in *Drosophila* by independent research groups.

(4) Line 546: add the Caudal Won-Jae Lee paper to state the posterior midgut is less microbicidal.

We added the reference at the right place, mentioning as well that it concerns adults.

(5) Figure 6 indicates what the cells are, shown by the arrow.

The sentence ‘the arrows point to TARMs’ is present in the legend of Fig6.

(6) Does the sphincter closure depend on hemocytes?

As mentioned above, the cells we identify as *TrpA1+* in the midgut junction may be the same cells described by Zaidman et al., 2012, and earlier by Lajeunesse et al., 2010. Inactivating hemocytes using the *Hml-Gal4* driver may also affect these *Dh31+* cells, as they share similarities with hemocytes, as pointed out by Zaidman et al. However, distinguishing between hemocytes and *Dh31+*/*TrpA1+* cells would require a genetic intersectional approach, which is beyond the scope of our current study.

Nevertheless, the possibility that these cells play a dual role in immunity (through blockage) and share characteristics with hemocytes while functioning as enteroendocrine cells (EECs) is quite intriguing and deserves further exploration in future studies.